# Improvements in the Resistance of the Banana Species to Fusarium Wilt: A Systematic Review of Methods and Perspectives

**DOI:** 10.3390/jof7040249

**Published:** 2021-03-25

**Authors:** Anelita de Jesus Rocha, Julianna Matos da Silva Soares, Fernanda dos Santos Nascimento, Adriadna Souza Santos, Vanusia Batista de Oliveira Amorim, Claudia Fortes Ferreira, Fernando Haddad, Janay Almeida dos Santos-Serejo, Edson Perito Amorim

**Affiliations:** 1Department of Biological Sciences, State University of Feira de Santana, Feira de Santana 44036-900, Bahia, Brazil; anelitarocha@gmail.com (A.d.J.R.); juliannamatos91@gmail.com (J.M.d.S.S.); feel.20@hotmail.com (F.d.S.N.); 2Bahia Education Secretary, Salvador 41745-004, Bahia, Brazil; adriadna_souza@yahoo.com.br; 3Embrapa Cassava and Fruit, Cruz das Almas 44380-000, Bahia, Brazil; vanusiaamorim50@gmail.com (V.B.d.O.A.); claudia.ferreira@embrapa.br (C.F.F.); fernando.haddad@embrapa.br (F.H.); janay.serejo@embrapa.br (J.A.d.S.-S.)

**Keywords:** *Musa* spp., *Fusarium oxysporum* f. sp. *cubense*, genetic improvement, resistance, state-of-the-art

## Abstract

The fungus *Fusarium oxysporum* f. sp. *cubense* (FOC), tropical race 4 (TR4), causes *Fusarium* wilt of banana, a pandemic that has threatened the cultivation and export trade of this fruit. This article presents the first systematic review of studies conducted in the last 10 years on the resistance of *Musa* spp. to *Fusarium* wilt. We evaluated articles deposited in different academic databases, using a standardized search string and predefined inclusion and exclusion criteria. We note that the information on the sequencing of the *Musa* sp. genome is certainly a source for obtaining resistant cultivars, mainly by evaluating the banana transcriptome data after infection with FOC. We also showed that there are sources of resistance to FOC race 1 (R1) and FOC TR4 in banana germplasms and that these data are the basis for obtaining resistant cultivars, although the published data are still scarce. In contrast, the transgenics approach has been adopted frequently. We propose harmonizing methods and protocols to facilitate the comparison of information obtained in different research centers and efforts based on global cooperation to cope with the disease. Thus, we offer here a contribution that may facilitate and direct research towards the production of banana resistant to FOC.

## 1. Introduction

Dessert bananas and plantains are very popular fruits worldwide. In 2018, approximately 116 million tons of bananas and 40 million tons of plantains were produced [1]. In terms of exports, bananas are among the most traded fruits globally, with almost 23 million tons (except for plantains) exported in 2017, representing almost 20% of global production [1]. Approximately 11.3 million hectares are dedicated to banana and plantain production worldwide, and there are more than 1000 varieties produced and consumed locally [2]. The Cavendish banana, which accounts for about 47% of global production, is the most traded [2]. In African regions, plantains comprise a significant and essential component, contributing considerably to food security and income generation for more than 70 million Africans [3,4,5]. Similarly, in Latin America and the Caribbean, 62% of total banana and plantain production (20 million tons) is consumed locally, and approximately 6.8 million tons of plantains are produced, of which 72% are traded on international markets, indicating the enormous importance of these crops for local food and food security throughout the region [6,7].

Among banana improvement programs’ objectives are achieving cultivars resistant to abiotic stressors, such as salinity [8,9] and drought [10,11,12]. Another major challenge for the global production of Musaceae species is the development of cultivars resistant to biotic stressors, represented by their primary pests, the banana root borer (*Cosmopolites sordidus*) and the nematodes *Meloidogyne* spp., *Pratylenchus coffeae*, and *Radopholus similis* [13,14,15,16,17,18,19], and disease-causing pathogens, including banana bunchy top virus (BBTV) [20,21], *Xanthomonas vasicola* pv. *musacearum* causing bacterial wilt [22,23,24,25], *Pseudocercospora fijiensis* causing black Sigatoka [26,27,28,29], and *Fusarium oxysporum* f. sp. *cubense* (FOC) causing *Fusarium* wilt [30,31,32,33].

FOC is one of the main biotic stress factors affecting bananas and *Fusarium* wilt is considered the most destructive and widely spread disease in the banana-producing regions around the world [34,35]. The causal agent is a soilborne fungus apparently considered hemibiotrophic; therefore, it initially establishes in a biotrophic relationship interacting with live plant cells of the host, and then in its necrotrophic phase, the host’s tissues are dead [30]. Frequently FOC persists in cultivated areas for years due to it is survival phase when it then interacts as saprophytic in cultural remains or produces resting spores known as chlamydospores besides surviving and multiplying in alternative hosts [30,35,36]. The disease is characterized by yellowing of the young leaves, and pseudostem splitting, and eventually death of the plant [30,37,38].

*Fusarium* wilt epidemics caused by race 1 (FOC R1), which occurred in Central America, caused the devastation of the susceptible “Gros Michel” cultivar plantations and was one of the most severe in the history of the crop in the Americas. For this reason, Gros Michel was replaced by cultivars of the subgroup Cavendish that are resistant to FOC R1 [39,40,41]. However, in the late 1980s, a highly virulent strain of FOC-infected Cavendish cultivars and spread to Asia, Africa, Indonesia, and more recently to South America [30,41,42]. Currently, *Fusarium* wilt can be considered a pandemic disease because of the spread of the tropical race 4 (FOC TR4) strain [43,44].

Chemical control is unfeasible and minimally effective, and it can be harmful to human health and the environment. Although still in its initial stages, biological control demonstrates promising results [31,45]. Low efficacy of the biological control is attributed to inherent factors to the dynamics of the disease’s primary inoculum, especially production of chlamydospores, which persists in cultivated areas, such as the capacity to survive in crop remains as an endophytic fungus in alternative hosts [30,36,46]. In addition, the genetic variability of the pathogen, resulting in new strains capable of infecting resistant cultivars, is another factor that limits the use of methods of disease management and control [47,48,49]. Therefore, efforts on the genetic improvement to achieve resistance to FOC R1 and FOC TR4 have been focused on finding resistant cultivars through traditional methods of germplasm selection or the generation of new cultivars by hybridization, genetic transformation, somaclonal variation, or mutation induction [31,50,51].

Until now, there are reviews available in the literature about *Fusarium* wilt related to epidemiology and disease management [30,35,41,52,53,54], biological control [45,55], genetic breeding for resistance [56,57] and one review about genomic aspects of *Musa* spp. for stress resistance [58].

The systematic reviews were mainly developed because of the need for rapid responses to human health issues, and nowadays, this tool has contributed to several study areas [59,60,61]. However, to our best knowledge, no systematic reviews on the genetic improvement of *Musa* spp. to resist Fusarium wilt have been published; only studies related to water stress in *Musa* spp. [62] and banana consumption [63]. Therefore, to provide detailed information on the subject and to collaborate with the information gathered so far, we propose a systematic approach to the studies on *Musa* spp., with a focus on genetic improvement for resistance to the FOC pathogen, through a systematic review of studies conducted over the last 10 years.

## 2. Materials and Methods

The free software State of the Art by Systematic Review (StArt) v.3.3 beta 03, developed by the Federal University of São Carlos (UFSCar), was used to perform a systematic review. This tool offers systematized answers to questions directed toward the objective of the review. The review process was performed in three stages—planning, execution, and summarization—according to the review flowchart in Figure 1, which followed the model proposed by Santos et al. [62].

### 2.1. Planning

A protocol to be followed during the review process was formulated, in which the title, objective, keywords, research questions, research sources, and inclusion/exclusion criteria of articles were defined during their selection and extraction. The StArt protocol is available for download at https://doi.org/10.5281/zenodo.4555385 (accessed on 22 February 2021). The research questions of the review are listed in Table 1.

To answer question 7, when there was no mention in the text of the location where the study was conducted, the search criteria within the article were standardized to the corresponding author’s mailing address to obtain information from which country the studies originated.

### 2.2. Execution: Search

Electronic surveys were conducted on the following databases, aiming to identify publications made available between January 2010 and December 2020: Scopus (http://www.scopus.com/), Web of Science (http://apps.isiknowledge.com), PubMed Central (https://www.ncbi.nlm.nih.gov/pmc/about/intro/), Springer (https://www.springer.com/br); Coordination for the Improvement of Higher Education Personnel Portal Journal (http://www.periodicos.capes.gov.br/), and Google Scholar (https://scholar.google.com.br/schhp?hl=en&as_sdt=0,5), using a standardized search string with the following keywords: *Musa* spp. and bananas and plantains and *Fusarium* wilt or *Fusarium oxysporum* f. sp. cubense or Panama disease and genetic resistance and markers and genes. This set of terms was used for research in all fields within the articles. The Boolean operators AND and OR were used to differentiate the search terms. Search results in each base were imported into BIBTEX, MEDLINE, or RIS formats, compatible with StArt. Relevant documents not found or published after the selection stage started were added manually. We did not consider using the name *Fusarium odoratissimum* proposed by Maryani et al. [64] in our standardized search due to the low number of published articles using this new suggested nomenclature, and this would limit the number of recovered articles in the database.

### 2.3. Execution: Selection and Extraction

In the selection stage, the articles that contained the terms adopted in the search string in the title, abstract, or keywords were accepted. In the extraction stage, where the number of articles was restricted, a single criterion to include articles was adopted, as follows: (I) articles that answer the protocol’s questions (Table 1). The criteria used to exclude articles in the extraction stage were (E) review articles, (E) theses, dissertations, manuals, and book chapters, (E) articles outside the subject, (E) articles published in event annals, (E) articles on genetic diversity of FOC, (E) articles on disease management strategies, and (E) articles on first reports of FOC. These criteria were considered to restrict the selected articles to the focus of this review since they do not answer the proposed questions about improving the resistance of *Musa* spp. to FOC. The preferred reporting items for systematic reviews and meta-analyses (PRISMA) checklist is presented for download at https://doi.org/10.5281/zenodo.4313617 (accessed on 9 December 2020).

### 2.4. Analysis of the Articles

The process of analyzing the articles was based on the calculation of the frequencies of articles related to each of the research questions. Subsequently, graphs, word clouds, and tables were prepared.

## 3. Results

### 3.1. Screening of Studies

The article screening process is represented by the flow chart in Figure 2. PubMed Central contributed the largest number of articles to this review, with 806 (50%) of the total, followed by Web of Science with 361 (22%) and Google Scholar with 319 (20%). The other databases, namely Scopus, Springer, and Coordination for the Improvement of Higher Education Personnel Portal Journal, contributed 69 (4%), 26 (2%), and 8 (0.5%) articles, respectively. Moreover, 22 (1.2%) articles that were not obtained automatically were added manually (Figure 2).

We identified 1612 articles from the database tracking, of which 234 were duplicated, and 1377 were eliminated in the selection process by reading the title, abstracts, and keywords, which did not fit the purposes of the research. In the extraction stage, 308 articles were analyzed. After reading the articles entirely, 213 were eliminated; hence, 95 were selected to compose the systematic review (Figure 2). The articles selected to compose the systematic review are available for consultation and download at https://doi.org/10.5281/zenodo.4555343 (accessed on 22 February 2021) and its origin and database in Appendix A.

A word cloud was generated from the keywords of the 95 articles for this review. As expected, there was a predominance of the articles with the keywords, “Fusarium oxysporum f. sp. cubense”, “Fusarium wilt”, “Musa”, “banana”, “disease”, “resistance”, “race”, tropical, and TR4 (Figure 3). Other keywords that had a remarkable frequency in the word cloud were “gene”, “Panama”, “transformation or transgenesis”, “plant”, “infection, “green fluorescent protein (GFP)”, “protein”, “SCAR”, “Acuminata”, “species”, “Cavendish”, and “polymerase chain reaction (PCR)” (Figure 3).

### 3.2. Known Origin Sites

Among the 95 articles, 53% were from China, followed by India (15%), Australia (12%), Brazil (7%), Malaysia (5%), Indonesia (4%), Uganda (4%), and other countries with a contribution of 1% (Figure 4). In the selected articles, 10 improvement programs located in different countries were mentioned, containing information with the potential for genetic improvement for the resistance of *Musa* spp. to *Fusarium* wilt (Figure 4).

Among the improvement programs cited, those that worked with crossbreeding to develop resistant cultivars were as follows: Honduras Foundation for Agricultural Research (Fundación Hondureña de Investigación Agrícola—FHIA), located in Honduras; Centre Africain de Recherches sur Bananiers et Plantains (CARBAP) in Cameroon; the International Institute of Tropical Agriculture (IITA) in Nigeria and Uganda; National Improvement Program of the Brazilian Agricultural Research Corporation (EMBRAPA) in Brazil; National Banana Research Center (NCRB) in India; National Research Organization (NARO) in Uganda; and Centre de Coopération Internationale en Recherche Agronomique pour le Développement (CIRAD) in Guadeloupe, French Antilles (Figure 4). In contrast, the improvement programs Taiwan Banana Research Institute (TBRI) and Guangdong Academy of Agricultural Sciences (GDAAS) in China worked with somaclonal variants and biotechnology.

### 3.3. Main Methods and Tools

Concerning the fungal races, the highest number of articles addressed specific studies with FOC TR4 (57%), 25.8% of the studies dealt only with FOC R1, and 10.1% of the articles performed comparative studies between FOC TR4 and FOC R1 (Figure 5A). Other studies with lower numbers conducted studies on subtropical race 4 (FOC STR4) (3.2%), FOC subtropical race 4 (STR4) and Foc TR4 (2.2%), and FOC R1 and FOC STR4 (1%) (Figure 5A). The highest frequency of articles was related to in silico (42.1%) and in vitro (32.6%) studies, followed by studies performed only in the greenhouse (12.6%), in the greenhouse and the field (5.3%), in the field only (4.2%), in the glasshouse (2.1%), and in other places (1%) (Figure 5B).

To evaluate *Fusarium* wilt symptoms, 26 scales were cited, divided among rhizome-discoloration symptoms, leaf-yellowing symptoms, and pseudostem division (Table 2). We found that 37% (*n* = 36) of studies adopted a scoring scale for external or internal *Fusarium* wilt symptoms (Table 2). According to the highest frequency of articles, the most-used scoring grades were from 1 to 6 for rhizome-discoloration and leaf-yellowing symptoms and from 1 to 3 for pseudostem division (Table 2). The most frequently cited scales were those of [65,66,67,68].

Among the main methods used for obtainment or characterization of plants resistant to *Fusarium* wilt, gene expression analysis represented 33% of the selected articles, followed by transgenesis (16%), symptomatology (13%), and resistance induction (11%) (Figure 6). In related articles, the other methods were classified as molecular markers (5%), symptomatology associated with the agronomic characterization of banana genotypes (5%), in vitro mutagenesis (4%), enzyme activity (3%), protein analysis and expression (3%), hybridization by crossbreeding (2%), and methods of somaclonal variation, clone selection, and somatic embryogenesis, each with a 1% frequency (Figure 6).

Among the tools used for the analysis and characterization of plants resistant to *Fusarium* wilt, the frequency of articles that employed analysis of reverse transcription-PCR (RT–qPCR) and PCR was the highest (35%). Analyses using bioinformatics tools were in 23% of the articles, and tissue culture represented 13% (Figure 7). Other tools adopted included the genetic transformation of the fungus with the GFP gene, the infection process by FOC strains (7%), banana transcriptome (7%), and phylogenetic analysis (7%). In addition to these tools, there was also a portion of articles using histochemistry and/or histology (6%) and other tools with a lower frequency (Figure 7).

Some articles used molecular markers associated with wilting resistance: Silva et al. [109], Wang et al. [110], and Wang et al. [51]. Among the markers associated with the resistance to FOC TR4, seven were from sequence characterized amplified region (SCAR)-type. One marker was associated with the susceptibility to FOC R1 [111] (Table 3). One random amplified polymorphic DNA (RAPD) molecular marker associated with the resistance to FOC R1 was found by Ghag et al. [98] (Table 3).

### 3.4. Resistance Sources

In the set of selected articles, many sources of resistance to *Fusarium* wilt were found for different FOC races (Table 4). Of the sources reported as resistant, 38% were triploid (AAA genome), 33% were diploid (AA genome), 12% were triploid (AAB genome), and 8% were tetraploid (AAAB genome); other genomes reported had a frequency of less than 5% (Figure 8 and Table 4).

The resistance sources reported that are exclusively related to FOC TR4 included the diploid cultivars, Pahang, Calcutta-4, Zebrina, Pisang Lilin, Malaccensis, Jari Buaya, and Tuu Gia, all with a higher frequency, according to the word cloud (Figure 9A). Besides these, other cultivars have also been reported as resistant to FOC TR4 in field tests, such as the hybrids FHIA-01, FHIA-02, SH-3748, SH-3362, FHIA-25, SH-3142, and SH-3362 (Figure 9A). According to genome frequency data related to resistance sources, most genotypes reported as resistant to FOC TR4 are AA diploid genomes (45%), AAA triploid genomes (21%), and AAB triploid genomes (18%) (Figure 9B).

### 3.5. Gene Expression Analysis

Figure 10A shows the gene categories present in studies on gene analysis and expression. The highest frequency of articles found genes associated with pathogenesis and defense (57%) (Figure 10A). Other genes, studied at a lower frequency, are related to RNAs (12%), hormone biosynthesis (10%), kinases (9%), transcription factors (6%), genes related to autophagy (4%), and starch biosynthesis (2%). A summary of the main genes related to each category can be found in Appendix A.

Methods for host plant inoculation to analyze gene expression after FOC infection are not standardized among the analyzed articles (*n* = 27), with several methods adopted (Figure 10B). The highest frequency of articles related to the inoculation method with conidia suspension at a concentration of 1 × 10^6^ spores mL^−1^ (38%), followed by the inoculation method by mechanical damage to the roots and then immersion in suspension at a concentration of 1 × 10^6^ mL^−1^ spores (19%). Other methods that were present in a single article represented 15% cumulatively (Figure 10B). The method of mechanical root damage and immersion in suspension at a concentration of 5 × 10^2^ spores mL^−1^ represented 12% of the articles and the methods of mechanical root damage and immersion in suspension at a concentration of 5 × 10^6^ spores mL^−1^ and mechanical root damage and soil infestation with 50 g of colonized millet seeds represented 8% of the articles (Figure 10B). Therefore, we observed that the main differences were related to whether the roots were wounded and the spore concentration adopted in each case regarding the inoculation method.

Table 5 is from the study by Wang et al. [115], which was modified, to show all the selected articles that evaluated the banana transcriptome infected by FOC TR4 and FOC R1. These studies observed the changes in expression of defense-related genes related to different enriched pathways, from gene annotation pathways, namely Gene Ontology (GO) annotation and the Kyoto encyclopedia of genes and genomes-based pathway analysis (KEGG-PATH). According to most transcriptome studies, the pathways activated after FOC infection were related to phenylpropanoid biosynthesis, sugar biosynthesis, cell wall modifications, flavonoid biosynthesis, and plant hormone signal transduction (Table 5). The main genes related to the above-mentioned pathways are listed in Appendix A.

### 3.6. Studies on the Achievement and Evaluation of Hybrids and on Genetic Inheritance of Musa *spp*.

The studies related to crossbreeding to obtain resistant hybrids or those focused on evaluating the genetic inheritance in *Musa* spp., as well as the parental lineages used and their genealogies, are listed in Table 6. Ssali et al. [93] produced hybrids from crossbreeding the resistant diploid TMB2X8075 (originated from the cross between SH3362 (AA) and Calcutta 4 (AA)) and Sukali Ndizi (AAB), which is also resistant to FOC R1 and 4, to evaluate the inheritance of the resistance of *Musa* spp. to FOC R1 in three F2 populations. Concerning the progeny, the authors found that 115 were susceptible, and 48 were resistant. Similarly, in the study by Arinaitwe et al. [31], crossbreeding between Monyet (*Musa acuminata* ssp. *Zebrina*) and Kokopo (*Musa acuminata* ssp. *Banksii*) were performed to identify suitable banana germplasm to generate a segregating F1 population and to understand the mode of inheritance of resistance to FOC R1 (Table 6).

The study by Ahmad et al. [108] is the first report of the genetic basis of resistance to FOC R1 in bananas using heterozygous wild banana *Musa acuminata* ssp. *malaccensis* (AA) to generate a mapping population and investigate the inheritance of resistance to FOC R1 and FOC TR4 through genetic mapping. This study demonstrated that resistance to FOC R1 is inherited as a single gene and that *M. acuminata* ssp. *malaccensis* is fertile and can be a potential parent to create resistance to *Fusarium* wilt.

Among the hybrids studied by Gonçalvez et al. [33], the improved diploids (CNPMF0038, CNPMF0513, CNPMF0767, and CNPMF1171) and the tetraploid hybrid BRS Princesa were considered moderately resistant (Table 6). All other hybrids evaluated in their study were considered resistant to *Fusarium* wilt caused by FOC R1. Gonçalvez et al. [33] mostly used improved male diploid parents, resulting from crossbreeding with diploids resistant to FOC R1 and FOC TR4, such as Calcutta 4 and M53.

### 3.7. Transgenesis

In the articles reporting the use of transgenesis (*n* = 14), the tool for genetic transformation was mediated by *Agrobacterium tumefaciens*, using embryogenic cell suspension culture. One exception is a method proposed by Subramaniam et al. [122], who, in addition to agroinoculation, developed a biolistics method. In this study, we used a table developed by Poon and Teo [123] as a model to show information about the works of this systematic review related to transgenesis (Table 7). In Table 7, we observed that most studies used the *Rasthali* cultivar (AAB) for genetic engineering.

Among the genes used for transgenesis, there was a considerable frequency of studies using transgenes as the antiapoptosis gene (Ced9) from the nematode *Caenorhabditis elegans* (Table 7). Two genes derived from *Musa acuminata* ssp. *malaccensis*, one related to pathogenesis (MaPR-10) and the other a resistance analog (RGA2), were also successfully used in this case as cisgenes. Other cell death genes, MusaDAD1, MusaBAG1, and MusaBI1, from *Musa acuminata* were also efficient, particularly MusaBAG1. In addition to these, the RNA interference technology enables the silencing of vital genes of FOC when employing small interfering RNA (siRNA) and intron-containing hairpin RNA (ihpRNA) (Table 7).

Four antimicrobial peptides from the plant species *Capsicum annuum*, *Petunia hybrida, Allium cepa*, and *Stellaria media* and an antifungal activity gene from *Trichoderma harzianum* were also successfully used to obtain resistant transgenic banana plants (Table 7).

### 3.8. Induction of Resistance

Among the inducers, the biocontrol agents *Bacillus subtilis*, *Trichoderma* spp., and *Penicillium citrinum* were the most reported for exploring induction of systemic resistance (Table 8). Chemical induction agents were also reported, such as the plant hormones abscisic acid (ABA), methyl jasmonate (MeJA), and salicylic acid (SA), in addition to benzothiadiazole (BTH). Other studies explored induced systemic resistance with the FOC pathogen in different ways (Table 8).

## 4. Discussion

### 4.1. Screening of the Studies

The studies analyzed were restricted to genetic improvement and in line with the questions proposed in the protocol formulated for this review. For this reason, articles on FOC genetic diversity, specific management strategies, and first reports of the disease were not considered in the analyses (Appendix A). Literature reviews were also excluded from the research to avoid bias since they could overestimate data, as many articles would be repeated.

Thus, the exclusion criteria used during the extraction stage of the article screening process revealed that many specific studies on FOC genetic diversity (*n* = 72) were generated in the last ten years, as well as many articles that escaped the proposed subject of this review (*n* = 47) and several literature reviews (*n* = 26) (Appendix A). Although these papers were excluded by the criteria, they revealed important aspects of the direction of research on *Fusarium* wilt in the last 10 years, considering the search string used.

The considerable amount of data on the genetic diversity of FOC generated in recent years was primarily in response to the need to understand the population structure of the pathogen in different locations and the evolutionary mechanisms of the fungus that culminate in the emergence of new races [48,136,137]. In fact, the potential for public investment in research that addresses the dissemination of the FOC TR4 can generate high returns and substantially delay the spread of this disease [138].

### 4.2. Locations of Knowledge Generation

A substantial amount of data on banana genetic improvement for resistance to *Fusarium* wilt was evaluated in this systematic review, the majority (50%) from China. This is a consequence of the number of projects to control *Fusarium* wilt in China, involving institutions, such as the Guangdong Academy of Agricultural Sciences (GDAAS), Chinese Academy of Tropical Agricultural Sciences (CATAS), Fujian Agriculture and Forestry University, Hainan Academy of Agricultural Sciences, and Guangzhou Institute of Agricultural Sciences [139]. In addition, China is among the countries in Southwest Asia where banana plants were domesticated [140], in which bananas are one of the fruits with the oldest consumption record, and the country that ranks second among the top 10 banana producers worldwide [2].

Besides China, India (16.7%), Australia (10.4%), and Brazil (7.3%) have also contributed to the improvement in research on *Musa* spp. India is the largest banana producer globally, whereas Brazil ranks fourth among banana producers [2]. Furthermore, these countries host important research institutions, which work on banana improvement for the development of resistant hybrids from germplasm collections, such as the Brazilian collections of the National Research Center for Banana (NRCB) and EMBRAPA. Australia was the first country to report and describe *Fusarium* wilt and one of the first countries facing major problems with FOC TR4, which led to the end of the Cavendish banana industry in the Northern Territory in 2015 [35,141,142].

Overall, in recent years, a major stimulus for the growth in studies on *Fusarium* wilt has been the emergence of FOC TR4 as the most devastating threat to bananas worldwide. A clear demonstration of this is the estimate that 17% of the current banana cultivation area, with an annual production of 36 million tons worth approximately US $10 billion at current prices, could be lost over the next 20 years because of *Fusarium* wilt, which would necessitate investment in research aimed at improving the crop in this scenario [138,143].

### 4.3. Gene Expression Analysis

The genome of the diploid species *Musa acuminata* ssp. *malaccensis*, which is the ancestor of most banana triploid cultivars, has been sequenced [144]. In the present study, 50% of the articles used the banana genome as a tool for analysis. For this reason, the highest frequency of articles was related to in silico studies (45%), as part of gene expression analyses (30%) that mostly performed RT–qPCR and PCR analyses (34%), as well as bioinformatic analyses (23%).

Identifying genes related to host defense is one of the first steps to understand the underlying mechanism of resistance to diseases in plants [145]. Concerning FOC, knowledge of global gene expression patterns, influenced by infection of different races, has enhanced our understanding of host responses to infection. Moreover, the availability of banana transcriptomes was highly useful to improve the annotation of the banana genome and for biological research [37]. Based on the knowledge of the global patterns of gene expression influenced by infection of FOC R1 and FOC TR4, Li et al. [37] found a large number of simple nucleotide polymorphisms (SNPs) and short insertions and deletions (indels), which previously had not been annotated in the Musa genome. Other transcriptomic studies observed the regulated expression of defense genes, cell wall-modifying genes, and a phytoalexin, flavonoids, lignin biosynthesis genes and jasmonic acid and other plant hormones and transcription factors [37,113,114,115,116,117,146,147].

The lack of standardization pertaining to the inoculation methods for evaluating gene expression should be questioned to develop a universal method for plant host inoculation so that the results could be equated and compared. It should be noted that a striking difference between these methods is the generation of wounds in the roots before exposing them to a suspension with the fungus, which in fact does not reflect a similar situation in the field, except when there are interactions with other microorganisms in the soil, such as nematodes [107]. Therefore, we consider that the inoculation methods adopted in most studies should be reconsidered because they are primarily related to the mechanical opening of wounds made with sterile needles or crushing the roots. In addition, differences related to the concentration of spores in the infection process generated marked changes in the plant response to infection and in gene expression, especially when a high inoculation pressure is considered, such as at the concentration of 5 × 10^6^ spores mL^−1^ associated with wound generation. Consistent with this information, we know that *F. oxysporum* is considered a hemibiotrophic pathogen because it begins its infection cycle as a biotroph and later changes to a necrotroph and as the gene expression changes that occur in the roots. Host responses may be prioritized to the perception of the pathogen, preventing the penetration of the root tissue during the biotrophic stage, which would not be possible to notice in previously injured tissue [46,148]. Therefore, we suggest that a standardized method should be adopted regarding the inoculation method of host plants to verify gene expression, aimed mainly to simulate situations in the system of banana cultivation, considering the mechanisms of dissemination of FOC that usually occur by movement and deposition of contaminated soil [30].

### 4.4. Studies on Resistance Sources

Although available edible banana cultivars originate from *M. acuminata* (genome A) and *Musa balbisiana* (genome B), genome B has been associated with the best vigor and tolerance to biotic and abiotic stresses and is, therefore, a target for *Musa* spp. improvement programs [149,150]. The AAA triploid genomes frequently occurred when considering all resistance sources related to both FOC R1 and FOC TR4 (Figure 8). In the studies with FOC TR4, the highest frequency of resistant genotypes were related to AA diploid genomes (Figure 9). This demonstrates that FOC TR4-related resistance sources are still mostly composed of wild diploids that have not yet been exploited for triploid cultivars, as in FOC R1, which already has a large panel of resistant cultivars available.

Thus, we have shown that some wild relatives of edible bananas, such as *M. itinerans*, Pahang, Calcutta 4, DH Pahang and Tuu Gia (Figure 9A), are valuable resources of resistance genes to FOC TR4 [77]. These data continue to be reaffirmed based on recent RNA-seq analyses that revealed aspects of the key responses of the relative resistance of wild banana to FOC TR4, where it could be seen that many differentially expressed genes were found in the resistant wild relative *Musa acuminata* ssp. *Burmanicoides* compared to the susceptible cultivar “Brazilian (AAA)“ [147].

An example of banana resistant to FOC R1 and TR4 are the triploid banana referred to as East African Highland bananas (EAHB), which a recent study has revealed that Mchare and Matooke hybrids resistant to FOC R1 can replace susceptible cultivars in areas of production severely affected by the fungus and are important resources for the generation of resistant banana [112].

The genetic basis of resistance to FOC R1 in banana has been studied in three articles, of which Arinaitwe et al. [31] and Ahmad et al. [108] suggested that resistance to *Fusarium* wilt in *Musa* spp. is conditioned by a single dominant locus of resistance, contradicting Ssali et al. [93], who concluded that the gene was recessively inherited. However, the conclusions by Ahmad et al. [108] were based on genetic analyses that included mapping studies and not just segregation data based on phenotypic characters.

### 4.5. Main Methods and Tools Adopted

One of the most-used tools (12%), together with the symptomatological assessments to understand Musa × FOC interaction processes, is the genetic transformation of different FOC isolates with the GFP gene. This method allows researchers to follow the movements of the fungus within the tissues and compare the colonization path used by different FOC races [37,46,72,137,151]. A FOC STR4 strain, transformed with the GFP gene, was used to monitor the movement of the pathogen in two susceptible cultivars, Cavendish *Williams* (Musa AAA) and *Lady Finger* (Musa AAB) [46]. Those authors detected the presence of FOC on the roots, rhizome, and outer leaf sheaths of the pseudostem before the appearance of external symptoms. Another study using this method verified that, in some cases, the banana rhizome plays an important role as a barrier to the pathogen, preventing its migration to the rest of the plant [103].

The studies carried out in greenhouses corresponded to 13% of the articles, those in greenhouses and in the field to 5%, and those only in the field to 3%. The articles focused only on assessing *Fusarium* wilt symptoms are few since, overall, this type of evaluation is complementary to several other analyses as a safe phenotypic confirmation of resistance. Most of the evaluation methods cited are related to the quantification of the severity of *Fusarium* wilt by visual categorization of the cross-sections based on the level of discoloration of the vascular tissue of the rhizome and the pseudostem of the root tissue, according to the scales mentioned in Table 2 [65,66,67,71,77,94].

The greatest difference found between the rating scales adopted for analysis and confirmation of banana resistance to FOC is related to the scoring grades for the disease’s severity. A universal scoring scale should be adopted, especially for comparison purposes between studies from different banana research centers, to avoid discrepant results, for example, when evaluating hybrids resulting from crossbreeding, plants obtained by transgenesis, resistance-induction, or other methods.

Although there are few studies with somaclonal variation (1%), this is a tool that presents promising results. The Cavendish somaclone GCTCV-218 for commercial cultivation under the name of Formosona, generated in 2004 by the Taiwan Banana Research Institute, is known to be tolerant to FOC TR4 and two other somaclonal variants of Cavendish called GCTCV-53 and GCTCV-119 [152]. In a recent study, tests with these Cavendish banana somaclones in northern Mozambique revealed that GCTCV-119 was more resistant to FOC TR4, but GCTCV-218 produced better bunches [5]. Another recent study obtained, through different combinations of plant regulators in a culture medium, two somaclones of the cultivar Prata-Anã, namely T2-1 and T2-2, which presented resistance to FOC race 1 in a greenhouse, characterizing an important result for the banana cultivation in Brazil since the pathogen FOC R1is present in most banana plantations and this cultivar is preferred by Brazilian consumers [153].

In the articles analyzed, transgenesis was the most-used method (14%), followed by resistance induction (10%), hybridization (4%), in vitro mutagenesis (4%), and somaclonal variation, clone selection, and somatic embryogenesis (1%). Although the transgenic method has a limitation related to the production of embryogenic cell suspensions, a time-consuming process, some protocols have facilitated their implementation [127,154]. Among the cited protocols, the most-used have been proposed by Ganapathi et al. [154], which included the establishment of embryogenic cell cultures from thin sections of the shoot tip of cultivated Rasthali (AAB) banana cultivar in vitro and by Khanna et al. [155], which proposed transformation mediated by *Agrobacterium tumefaciens* assisted by centrifugation (CAAT) from male flower embryogenic cells suspensions of the Cavendish (AAA) and *Lady Finger* (AAB) cultivars. A protocol established by Yip et al. [69] proposes the substitution of embryogenic cell suspensions for meristematic tissue, where they use multiple shot clump (MSC) of Pei Chiao (AAA) and Gros Michel (AAA) bananas induced from shoots in the rhizome in MS medium; this could be another feasible option for banana cultivars where suspension cultures are difficult to establish. Another protocol was proposed by Subramaniam et al. [122] using the biobalistic gun method for the transformation of the ‘Rastali’ (AAB) banana cultivar. In addition, the availability of banana genes (cisgenes) and genes from other appropriate sources (transgenes) allowed the development and evaluation of transgenic plants (Table 6).

Conventional resistance improvement methods using hybridization between fertile diploids and crossbreeding with triploid or tetraploid cultivars are efficient. However, they have some limitations concerning the polyploid nature of the cultivars and the low female fertility, as well as the long life cycle leading to a long reproductive cycle [156,157]. Other challenges are related to the need for a large space, which results in high costs and limited knowledge about resistance genetics [31,158,159].

Transgenic methods permit the addition of a single gene or several genes to a highly desirable cultivar quickly [81,124,125,126]. Due to the sterility of these cultivars, the flow of transgenes and the crossing of modified genes between wild Musa species are unlikely; therefore, genetically modified (GM) bananas could be compatible with organic agriculture [159]. In addition, although no genome editing data associated with obtaining *Fusarium* wilt-resistant cultivars were identified in this study, the potential for using the CRISPR/Cas9, a genome-editing tool for the development of disease-resistant banana varieties, also has been reported. The use of genome editing (GE) with the availability of a whole-genome sequence and its potential applications to develop disease-resistant bananas opens new areas of research [160,161,162,163,164,165]. Although there are no published data in banana breeding, another potential tool to be applied is resistance gene enrichment sequencing (RenSeq), a technology that enables the discovery and annotation of pathogen resistance gene families in plant genome sequences. The use of this high-throughput technique was well demonstrated in wheat (*Triticum estivum*) [164] and potato (*Solanum tuberosum*) [165].

These data encourage discussions on the current status of biosafety regulations and laws on the marketing of GM products that face some challenges because of the regulation of these products in several countries [162,163]. Furthermore, their outlook indicated that investments in GM banana plantations would bring few beneficiaries, given the assumption that countries with export-oriented banana production would not adopt GM varieties because of political and consumer concerns [138]. In this sense, it seems reasonable to invest more in improvements based on crossbreeding, considering that there are sources of resistance to *Fusarium* wilt caused by FOC R1 and FOC TR4, which enables the selection of resistant hybrids within the progeny generated.

Using an ex-ante quantitative risk index model, Staver et al. [138] showed that investments in different research areas assessed to address the threat and projected losses from FOC TR4 would provide positive returns and contribute to a reduction in poverty. Moreover, there would be superior returns in poverty reduction, especially in Africa, in the face of investments in the research areas related to the conventional improvement of cultivars resistant to *Fusarium* wilt, as well as in the research area related to improving exclusion and surveillance, as well as measures to eradicate or contain the disease, with 850,000 and 807,000 people lifted out of poverty in each case, respectively.

### 4.6. Perspectives

In this study, we found that several articles in the last 10 years have focused on a variety of analyses to improve our understanding and identification of genetic, molecular, biochemical, or structural mechanisms of banana resistance to FOC, based on a set of tools. Based on these articles, we also showed that there are sources of resistance to FOC R1 and FOC TR4 in banana germplasms and that the data generated in these studies are the basis for obtaining cultivars resistant to *Fusarium* wilt. Moreover, they can contribute significantly to the expansion of resistant cultivars, including those for export. Although there is not yet a banana cultivar resistant to FOC TR4 that can replace the cultivars of the Cavendish subgroup, from the resistance sources found in different studies, it would be possible to develop a “type” similar to the Cavendish cultivars resistant to FOC TR4 or other races.

Concerning the improvement methods, there is a growing incentive for new precise and efficient genetic technologies, and the use of the CRISPR/Cas9 genome editing tool will also contribute to obtaining banana cultivars with FOC resistance in a short span of time. Other tools, which explore acquired and induced systemic resistance, also emerged as important means to achieve resistance to the pathogen, supported by experiments on tissue culture. Meanwhile, conventional improvement seeks to overcome the challenges inherent to the plant species by offering seemingly more appropriate measures with a focus on family-based agriculture of banana production systems worldwide. Nevertheless, the debate concerning various improvement methods should not be focused on just one method since all of them contribute to improving the crop, and the existence of different scenarios of banana production should be considered for the use of each method.

Furthermore, it is important to emphasize that the results obtained in this study are linked to the keywords used in the search string. The use of different terms could lead to the inclusion and exclusion of other articles in the systematic review and, consequently, lead to other methods and conclusions.

## 5. Conclusions

Improvement programs of *Musa* spp. have sought to reinforce their methods through new technologies and accumulate knowledge on resistance to *Fusarium* wilt. The genome sequencing of Musa is a widely used data source for improving the identification and analysis of resistance-related genes. The production of transgenic bananas has been explored, leading to the need for social exposure regarding the acceptance of such products. Although the use of genome editing tools, such as CRISPR/Cas9, to obtain resistance to *Fusarium* wilt in banana plants has not been performed, it is a method with promising prospects. In this review, we highlighted sources of resistance to FOC (R1 and TR4) based on diploids resistant to *Fusarium* wilt, which is the starting point for genetic improvement.

Therefore, we confirm that genetic improvement is the best strategy for combating *Fusarium* wilt by expanding resistant cultivars to producers. From the data collected in our systematic review, we believe that future research efforts can be based on integrating the knowledge obtained thus far to obtain results with greater applicability and direct the next steps in research to produce banana species resistant to *Fusarium* wilt. We suggest that future studies address the following questions: How can we exploit germplasm sources resistant to FOC R1 and FOC TR4 in improvement programs? Could the standardization of protocols for plant inoculation facilitate the comparison of data regarding gene expression analysis? Should a universal scoring scale contemplating the disease’s external and internal symptoms be elaborated based on existing scales? Can existing molecular markers be used in a standard-assisted selection protocol for resistance to FOC R1 and FOC TR4?

In addition, strategies based on the integration of knowledge from different *Musa* spp. improvement research centers should be adopted for cooperative efforts so that different improvement programs can cooperate on a global scale. Considering that the current banana export scenario is based exclusively on a single group, strategies should be considered to ensure the agribusiness export’s sustainability, prioritizing the production of other cultivars resistant to FOC.

## Figures and Tables

**Figure 1 jof-07-00249-f001:**
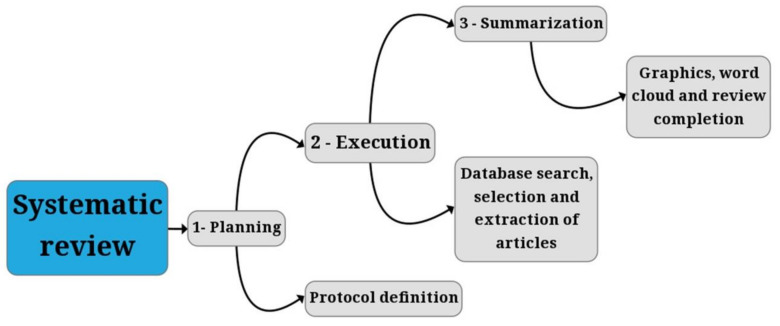
General systematic literature review flowchart. Source: author’s compilation.

**Figure 2 jof-07-00249-f002:**
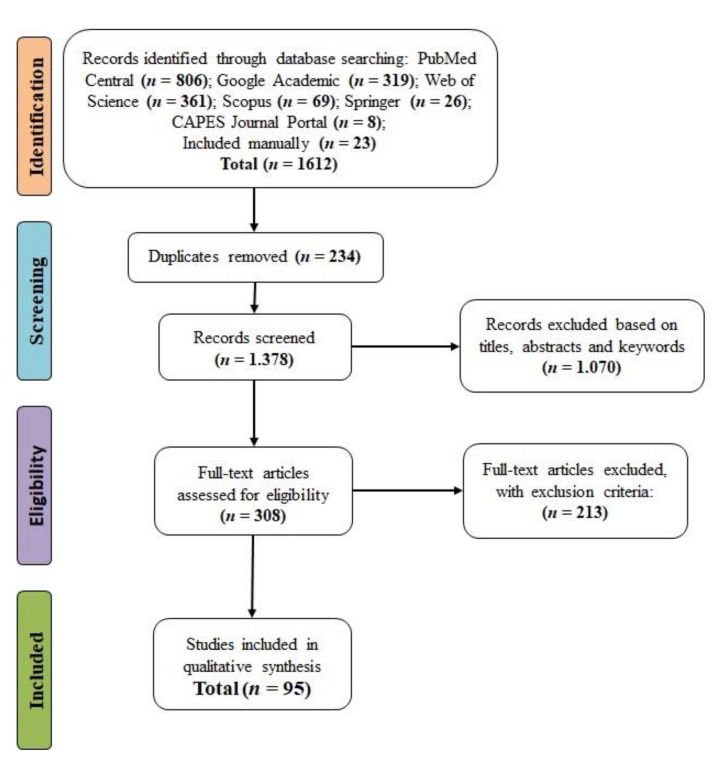
Flow diagram (preferred reporting items for systematic reviews and meta-analyses (PRISMA)). The selection process of studies for inclusion or exclusion in the systematic review on genetic improvement of banana for resistance to *Fusarium* wilt. *n* = number of studies. The flow diagram was based on a model available at http://prisma-statement.org/PRISMAStatement/FlowDiagram. CAPES: Coordination for the Improvement of Higher Education Personnel (accessed on 9 December 2020).

**Figure 3 jof-07-00249-f003:**
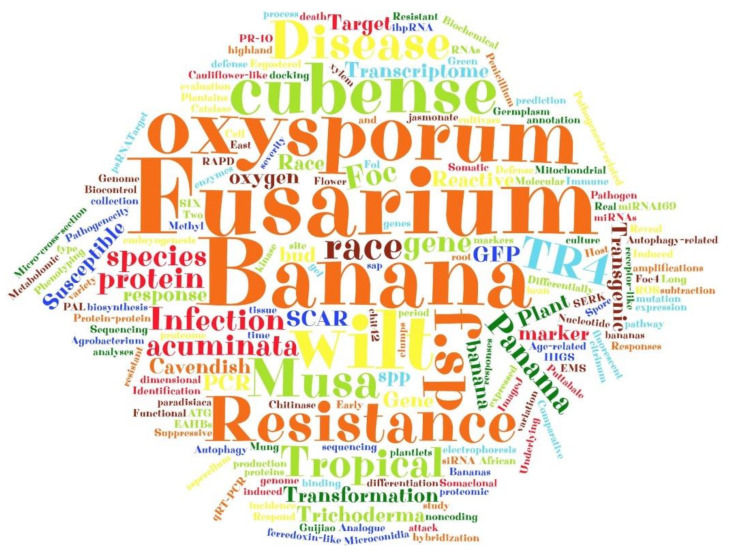
Word cloud generated from article keywords of selected articles to compose a systematic review on breeding *Musa* to *Fusarium* wilt. The word cloud was created in a free online generator (https://www.wordclouds.com/, accessed on 16 August 2020), based on the frequency of each keyword.

**Figure 4 jof-07-00249-f004:**
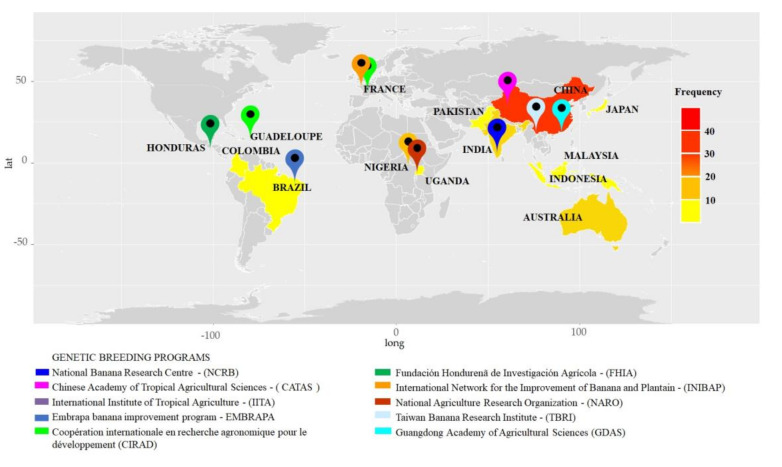
Frequency of articles on genetic improvement of *Musa* spp. to *Fusarium* wilt published in the last ten years in different countries and genetic breeding programs of the banana mentioned. The light yellow tones indicate a frequency below 10% of the articles considered in this review; the intermediate tones indicate frequencies between 10 and 30%, and the intense red tones indicate frequencies above 40%. The location icons indicate the locations of Musa breeding programs identified by the colors. The map was plotted in R, using the packages maps, ggmap, geosphere, Eurostat, GADMTools, country code and ggplot2. lat: latitude; long: longitude.

**Figure 5 jof-07-00249-f005:**
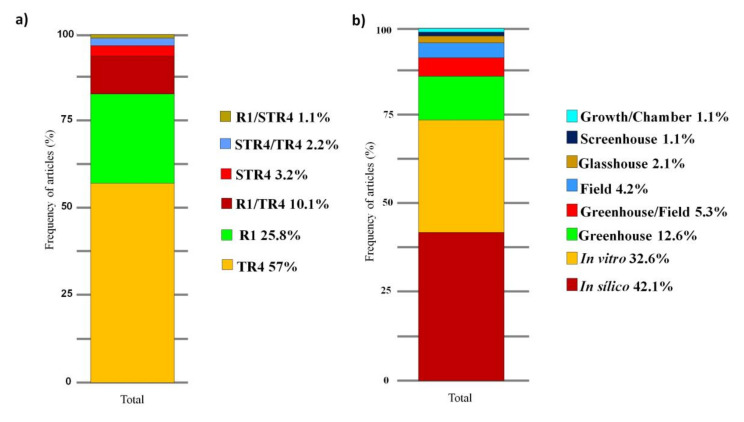
Stacked bar chart of the frequency of articles with different races of *Fusarium oxysporum* f. sp. *cubense* in the past ten years (**a**). Places of achievement of work in articles on the improvement of banana plants to *Fusarium* wilt carried out in the last 10 years (**b**). R1: race 1; STR4: subtropical race 4; TR4: tropical race 4.

**Figure 6 jof-07-00249-f006:**
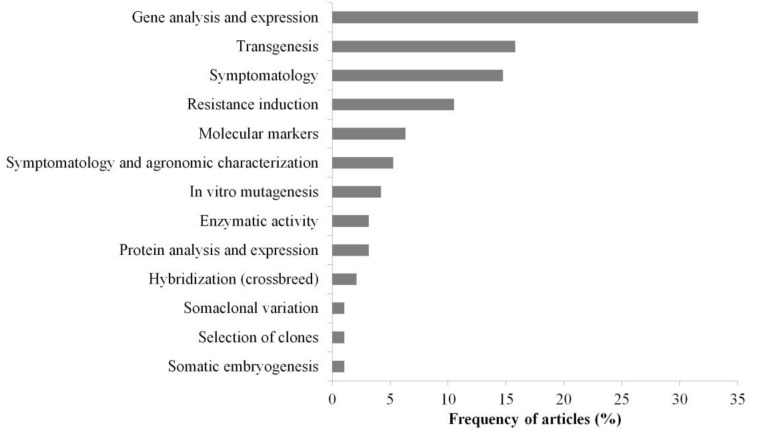
Banana plant breeding techniques used to supplant *Fusarium* wilt in articles published in the last 10 years.

**Figure 7 jof-07-00249-f007:**
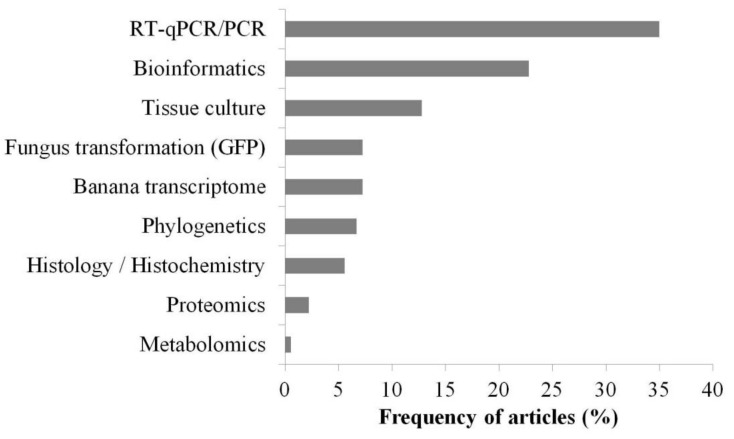
Frequency of articles associated with the main tools used in studies on banana plant breeding to *Fusarium* wilt in the last 10 years. The frequency considered that more than one tool was used per article. RT–qPCR/PCR: reverse transcription-PCR/polymerase chain reaction/ GFP: green fluorescent protein.

**Figure 8 jof-07-00249-f008:**
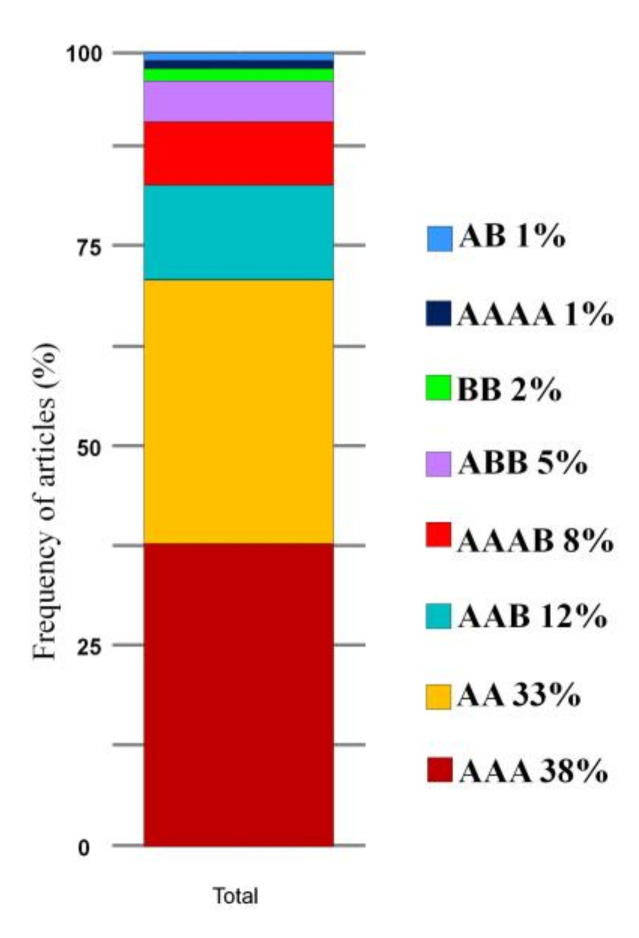
Frequency of genomes associated with sources of resistance to *Fusarium* wilt in studies on banana breeding carried out in the last ten years.

**Figure 9 jof-07-00249-f009:**
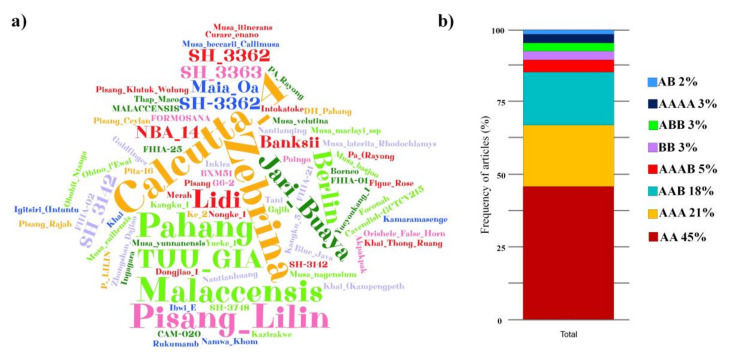
Sources of resistance to *Fusarium oxysporum* f. sp. *cubense* tropical race 4 (TR4) in studies on the improvement of banana plants to *Fusarium* wilt in the last ten years. (**a**) Word cloud of the frequency of cited sources of resistance. (**b**) Frequency of genomes related to the sources of resistance cited.

**Figure 10 jof-07-00249-f010:**
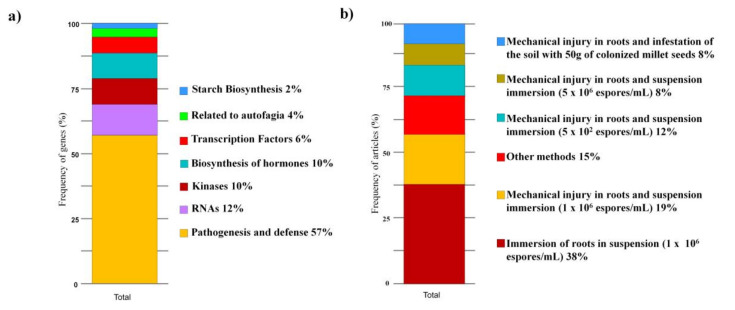
Gene expression studies of banana plants infected with *Fusarium oxysporum* f. sp. *cubense* in articles carried out in the last ten years. Categories of genes associated with the frequency of articles (**a**) and frequency of methods used for inoculation of plants to check gene expression (**b**).

**Table 1 jof-07-00249-t001:** List of questions about the genetic improvement of *Musa* for resistance to *Fusarium* wilt to be answered by a systematic review of studies carried out in the last ten years.

Research Questions
**Q1**: What are the known sources of resistance (germplasm) to *Fusarium oxysporum* f. sp. *cubense*?
**Q2**: Which breeding programs work on the resistance to *Fusarium* wilt with respect to cultivar development?
**Q3**: Which genes are reported associated with resistance to *Fusarium oxysporum* f. sp. *cubense* in *Musa* spp.
**Q4**: What breeding techniques are associated with overcome *Fusarium* wilt?
**Q5**: Which biotechnological tools are used for assisted selection for resistance to *Fusarium oxysporum* f. sp. *cubense*?
**Q6**: Which germplasm collections have information with the potential for genetic improvement to *Fusarium* wilt?
**Q7**: What is the frequency of studies by country, and which programs of improvement work with crossbreeding in order to develop resistant cultivars?
**Q8**: Are there scales to assess the disease? What is the difference between them?
**Q9**: How often is the banana genome used?

**Table 2 jof-07-00249-t002:** Scales of grades for assessing symptoms of *Fusarium oxysporum* f. sp. *cubense* reported in articles on banana breeding to *Fusarium* wilt conducted in the last ten years.

Article	Internal Symptoms	External Symptoms		Scale Reference
	Rhizome Discoloration	Yellowing of the Leaf	Pseudostem Division	
	Degrees of Scale			
Yip et al. [69]	0–3			[69] *
Orr et al. [70]	1–6			[71]
Chen et al. [72]	1–8			[68]
Warman and Aitken [46]	1–6			[66]
Baharum et al. [73]	1–8			[68]
Zhang et al. [74]	0–4	0–4		[75,76]
Zuo et al. [77]	1–5			[77] *
Ribeiro et al. [78]	0–5	0–4		[67,79]
Wei et al. [80]		0–4		[80] *
Garcez et al. [81]	0–5	0–5		[67,82]
Li et al. [75]	0–3	0–3		[75] *
Ghag et al. [83]		1–6		[66]
Smith et al. [84]	1–6			[65,85]
Mohandas et al. [86]	1–6	0–5		[65,87]
Ting et al. [88]		0–5		[88] *
Paul et al. [89]		1–5	1–3	[89] *
Sun et al. [90]	1–5	1–5		[64,91]
Wu et al. [92]		1–6		[92] **
Ssali et al. [93]		1–6		[94]
Li et al. [95]	0–4	0–5		[95] *
Ghag et al. [96]		1–6		[96] *
Saraswathi et al. [97]	1–5	1–5		[66,91]
Ghag et al. [98]		1–6		[83]
Sun et al. [76]		0–4		[76] *
Wu et al. [99]		1–6		[99] **
Magambo et al. [100]		1–5	1–3	[68]
Smith et al. [101]	1–6			[65]
García-Bastidas et al. [102]	1–6	1–4		[102] *
Arinaitwe et al. [31]	1–5	1–6	1–3	[71]
Cheng et al. [103]	1–8			[68]
Gonçalves et al. [33]	1–5	1–6		[67,104]
Buregyeya et al. [105]	1–6		1–3	[94]
Sunisha et al. [106]		1–5	1–3	[89]
Rocha et al. [107]	1–5	1–4		[104]
Ahmad et al. [108]	1–6	1–4		[102]

* use their own scale; ** in vitro.

**Table 3 jof-07-00249-t003:** Molecular markers associated with banana breeding strategies to *Fusarium* wilt in articles carried out in the last ten years.

Name	Type	Function	Citation
ScaU1001	SCAR	Resistance to FOC TR4	[109]
SuscPD	SCAR	Susceptibility to FOC 1	[111]
Lipoxygenase (gene)	RAPD	Resistance to FOC 1	[98]
ScaU1001	SCAR	Resistance to FOC TR4	[110]
ScaS0901			
SC1/SC2	SCAR	Resistance to FOC TR4	[51]
SC3/SC4			
SC5/SC6			
SC7/SC8			

SCAR: sequence characterized amplified region; RAPD: random amplified polymorphic DNA.

**Table 4 jof-07-00249-t004:** Sources of resistance to *Fusarium oxysporum* f. sp. *cubense* characterized in articles on the improvement of banana to *Fusarium* wilt carried out in the last ten years.

*Musa* Germplasm	*Musa* Genome	Race	Level of Tolerance or Resistance to Races	Institution and Location/Country Where Germplasm Was Screened	Known Use to Mitigate Fusarium Impact	References
M53	AA	Race 1	R	Embrapa cassava and fruit growing, Brazil	In breeding programs	[33,78,107]
Birmanie	AA	Race 1	R	Embrapa cassava and fruit growing, Brazil	In breeding programs	[78,107]
PA Songkla	AA	Race 1	R	Embrapa cassava and fruit growing, Brazil	In breeding programs	[78]
Pirua	AAA	Race 1	R	Embrapa cassava and fruit growing, Brazil	Brazil	[78]
Imperial	AAA	Race 1	R	Embrapa cassava and fruit growing, Brazil	Brazil	[78]
Poyo	AAA	Race 1	R	Embrapa cassava and fruit growing, Brazil [78], DAFF, Australia [84]	Brazil, Africa	[78,84]
BRS Vitória	AAAB	Race 1	R	Embrapa cassava and fruit growing, Brazil	Brazil	[107]
Ambei	AA	Race 1	R	Embrapa cassava and fruit growing, Brazil	In breeding programs	[78]
Walebo	AAA	Race 1	R	Embrapa cassava and fruit growing, Brazil	Brazil	[78]
Kongo FRF 1286	AAA	Race 1	R	Embrapa cassava and fruit growing, Brazil	Brazil	[78]
Pisang Nangka	AAB	Race 1	R	Embrapa cassava and fruit growing, Brazil	Brazil, Africa, Australia	[78]
Pisang Jaran	AA	Race 1	R	Embrapa cassava and fruit growing, Brazil	In breeding programs	[78]
Tjau Lagada	AA	Race 1	R	Embrapa cassava and fruit growing, Brazil [33,78]	In breeding programs	[33,78]
Mangana	AA	Race 1	R	Embrapa cassava and fruit growing, Brazil	In breeding programs	[78]
Pisang Pipit	AAA	Race 1	R	Embrapa cassava and fruit growing, Brazil	In breeding programs	[78]
Pisang Rojo Uter	AA	Race 1	R	Embrapa cassava and fruit growing, Brazil	In breeding programs	[78]
2803-01	AA	Race 1	R	Embrapa cassava and fruit growing, Brazil	In breeding programs	[78]
GN. P. Formoso	AAA	Race 1	R	Embrapa cassava and fruit growing, Brazil	In breeding programs	[109]
Pisang Tongat	AA	Race 1	R	Embrapa cassava and fruit growing, Brazil	In breeding programs	[78]
Mchare cultivars	AA	Race 1	R	Stellenbosch University, South Africa (Arusha, Tanzania)	Africa	[112]
Mchare hybrids	AA	Race 1	R	Stellenbosch University, South Africa (Arusha, Tanzania)	Africa	[112]
NARITA hybrids	AA	Race 1	R	Stellenbosch University, South Africa (Kawanda, Uganda)	Africa	[112]
Figo Cinza	ABB	Race 1	R	Embrapa cassava and fruit growing, Brazil [78] Banana Germplasm Bank of the Itajaí Research Station [111]	Brazil	[78,111]
M-61	AAA	Race 1	R	Embrapa cassava and fruit growing, Brazil	In breeding programs	[78]
Nanicão Magario	AAA	Race 1	R	Embrapa cassava and fruit growing, Brazil	Brazil	[78]
Buitenzorg	AA	Race 1	R	Embrapa cassava and fruit growing, Brazil	In breeding programs	[78]
BRS Platina	AAAB	Race 1	R	Embrapa cassava and fruit growing, Brazil [33,78,107] Itajaí Research Station, Brazil [111]	Brazil	[33,78,107,111]
Nanica	AAA	Race 1	R	Embrapa cassava and fruit growing, Brazil	Brazil	[78,107,109]
Pisang Ustrali	AAB	Race 1	R	Embrapa cassava and fruit growing, Brazil	In breeding programs	[78]
Markatooa	AAA	Race 1	R	Embrapa cassava and fruit growing, Brazil	In breeding programs	[78]
Robusta	AAA	Race 1	R	Embrapa cassava and fruit growing, Brazil	In breeding programs	[78]
BRS Pacovan Ken	AAAB	Race 1	R	Embrapa cassava and fruit growing, Brazil	Brazil	[78,107]
BRS Princesa	AAAB	Race 1	R	Federal Institute of the Triangulo Mineiro, Brazil [74], Embrapa cassava and fruit growing, Brazil [33,107]	Brazil	[33,81,107]
BRS Japira	AAAB	Race 1	R	Federal Institute of the Triangulo Mineiro, Brazil [81] Embrapa cassava and fruit growing, Brazil [107]	Brazil	[81,107]
BRS Tropical	AAAB	Race 1	R	Federal Institute of the Triangulo Mineiro, Brazil	Brazil	[81]
Grand Naine	AAA	Race 1	R	Embrapa cassava and fruit growing, Brazil [33,78,107,109], Federal University of Santa Catarina, Brazil [111]	Cavendish for export	[33,78,107,109,111]
Nanicão	AAA	Race 1	R	Embrapa cassava and fruit growing, Brazil [78,111], Federal University of Santa Catarina, Brazil [111]	Brazil	[78,109,111]
SCS452 Corupá	AAA	Race 1	R	Federal University of Santa Catarina, Brazil	Brazil	[111]
Zellig	AAA	Race 1	R	Federal University of Santa Catarina, Brazil	Brazil	[111]
Figo	ABB	Race 1	R	Embrapa cassava and fruit growing, Brazil [78], Federal University of Santa Catarina, Brazil [111]	In breeding programs	[78,111]
FHIA-17	AAAA	Race 1	R	DAFF, Australia	Honduras, Brazil	[84]
SH-3640.10	AAAB	Race 1	R	DAFF, Australia	Honduras, Brazil, Mozambique, Cameroon	[84]
Long Tavoy	*	Race 1	R	University of Malaya, Kuala Lumpur, Malaysia	In breeding programs	[31]
Kasaska	*	Race 1	R	University of Malaya, Kuala Lumpur, Malaysia	In breeding programs	[31]
Monyet	*	Race 1	R	University of Malaya, Kuala Lumpur, Malaysia	In breeding programs	[31]
Mwitu Pemba	*	Race 1	R	University of Malaya, Kuala Lumpur, Malaysia	In breeding programs	[31]
Hom Thong Mokho	AAA	Race 1	R	Department of Agriculture and Fisheries (DAF), Queensland, Australia	Australia	[101]
Mambee Thu	AA	Race 1	R	Embrapa cassava and fruit growing, Brazil	In breeding programs	[78]
PV03-79	AAAB	Race 1	R	Embrapa cassava and fruit growing, Brazil	In breeding programs	[78]
Terra Maranhão	AAB	Race 1	R	Embrapa cassava and fruit growing, Brazil	Brazil	[107]
Williams	AAA	Race 1	R	DAFF, Australia [101], Federal University of Santa Catarina, Brazil [111]	Cavendish for export	[101,111]
Williams	AAA	STR4	SS	University of Queensland, Australia	Cavendish for export	[72]
SH-3217	AA	STR4	R	University of Queensland, Australia	In breeding programs	[72]
Ma250	AA	STR4	R	University of Queensland, Australia	In breeding programs	[72]
Pisang Bangkahulu	AA	STR4	R	University of Queensland, Australia	In breeding programs	[72]
M61 Guadelope	*	STR4	SS	University of Queensland, Australia	In breeding programs	[72]
CAM-020	AAA	STR4	S	University of Queensland, Australia	In breeding programs	[72]
SH-3142	AA	TR4	SS	(IFTR-GDAAS), Guangzhou, China	In breeding programs	[77]
FHIA-1 (“Gold Finger”)	AAAB	TR4	S	GDAAS, Guangzhou, China	Australia, Brazil, Mexico, Colombia, EUA	[75]
GCTCV-119	AAA	TR4	HR	Guangdong Academy of Agricultural Sciences, Guangzhou, China	China, Taiwan, The Philippines, Mozambique.	[92]
M61 Guadeloupe	*	TR4	R	University of Queensland, Australia	In breeding programs	[72]
CAM-020	AAA	TR4	R	University of Queensland, Australia	In breeding programs	[72]
Ibwi E	AAA	TR4	R	(IFTR-GDAAS), Guangzhou, China	EAHBs	[77]
Igitsiri (Intuntu)	AAA	TR4	R	(IFTR-GDAAS), Guangzhou, China	EAHBs	[77]
Ingagara	AAA	TR4	R	(IFTR-GDAAS), Guangzhou, China	EAHBs	[77]
Inkira	AAA	TR4	R	(IFTR-GDAAS), Guangzhou, China	EAHBs	[77]
Intokatoke	AAA	TR4	R	(IFTR-GDAAS), Guangzhou, China	EAHBs	[77]
Kazirakwe	AAA	TR4	R	(IFTR-GDAAS), Guangzhou, China	EAHBs	[77]
Mbwazirume	AAA	TR4	HR	(IFTR-GDAAS), Guangzhou, China	Africa	[77]
Akpakpak	AAB	TR4	HR	(IFTR-GDAAS), Guangzhou, China	Africa	[77]
Curaré Enano	AAB	TR4	R	(IFTR-GDAAS), Guangzhou, China	Africa	[77]
Obino l’Ewai	AAB	TR4	R	(IFTR-GDAAS), Guangzhou, China	Africa	[77]
Obubit Ntanga	AAB	TR4	R	(IFTR-GDAAS), Guangzhou, China	Africa	[77]
Orishele False Horn	AAB	TR4	HR	(IFTR-GDAAS), Guangzhou, China	Africa	[77]
Pisang Ceylan	AAB	TR4	R	(IFTR-GDAAS), Guangzhou, China	Africa	[77]
Pisang Rajah	AAB	TR4	R	(IFTR-GDAAS), Guangzhou, China	Africa	[77]
*Musa itinerans*	*	TR4	HR	(IFTR-GDAAS), Guangzhou, China	In breeding programs	[77]
CIRAD930/DH Pahang	AA	TR4	HR	(IFTR-GDAAS), Guangzhou, China	In breeding programs	[77]
NBA 14	AA	TR4	R	(IFTR-GDAAS), Guangzhou, China	In breeding programs	[77]
Banksii	AA	TR4	R	(IFTR-GDAAS), Guangzhou, China	In breeding programs	[77]
Maia Oa	AA	TR4	R	(IFTR-GDAAS), Guangzhou, China	In breeding programs	[77]
Zebrina	AA	TR4	SS	(IFTR-GDAAS), Guangzhou, China	In breeding programs	[77]
Pa (Rayong)	AA	TR4	R	(IFTR-GDAAS), Guangzhou, China	In breeding programs	[77]
Figue Rose	AA	TR4	R	(IFTR-GDAAS), Guangzhou, China	In breeding programs	[77]
Khai (Kampengpeth)	AA	TR4	R	(IFTR-GDAAS), Guangzhou, China	In breeding programs	[77]
Tani	BB	TR4	R	(IFTR-GDAAS), Guangzhou, China	In breeding programs	[77]
Pisang Klutuk Wulung	BB	TR4	R	(IFTR-GDAAS), Guangzhou, China	In breeding programs	[77]
*Musa beccarii* Callimusa	*	TR4	R	(IFTR-GDAAS), Guangzhou, China	In breeding programs	[77]
*Musa laterita* Rhodochlamys	*	TR4	R	(IFTR-GDAAS), Guangzhou, China	In breeding programs,	[77]
*Musa maclayi* ssp.	*	TR4	R	(IFTR-GDAAS), Guangzhou, China	In breeding programs,	[77]
Khai Thong Ruang	AAA	TR4	R	(IFTR-GDAAS), Guangzhou, China	In breeding programs	[77]
Kamaramasenge	AB	TR4	R	(IFTR-GDAAS), Guangzhou, China	In breeding programs	[77]
Rukumamb	AAB	TR4	R	(IFTR-GDAAS), Guangzhou, China	Australia, Papua New Guinea	[77]
Thap Maeo	AAB	TR4	R	(IFTR-GDAAS), Guangzhou, China	Brazil, Honduras	[77]
Foconah	AAB	TR4	R	(IFTR-GDAAS), Guangzhou, China	In breeding programs	[77]
Poingo	AAB	TR4	R	(IFTR-GDAAS), Guangzhou, China	In breeding programs	[77]
FHIA-21	AAAB	TR4	R	(IFTR-GDAAS), Guangzhou, China [77], DAFF, Australia [84]	In breeding programs	[77,84]
Blue Java	ABB	TR4	R	(IFTR-GDAAS), Guangzhou, China [77], Embrapa cassava and fruit growing, Brazil [107]	China, Africa, Brazil	[77,107]
Namwa Khom	ABB	TR4	HR	(IFTR-GDAAS), Guangzhou, China [77], DAF, Australia [101]	China, Africa, Thailand	[77,101]
FHIA-02	AAAA	TR4	R	DAFF, Australia	Africa, Brazil, Colombia, Honduras	[72,84]
SH-3362 (“Pita-16”)	*	TR4	R	DAFF, Australia	In breeding programs	[72]
*M. yunnanensis*	*	TR4	R	South China Agricultural University	Wild germplasm	[75]
*M. basjoo*	*	TR4	R	South China Agricultural University	Wild germplasm	[75]
*M. nagensium*	*	TR4	R	South China Agricultural University	Wild germplasm	[75]
*M. ruiliensis*	*	TR4	R	South China Agricultural University	Wild germplasm	[75]
*M. velutina*	*	TR4	R	South China Agricultural University	Wild germplasm	[75]
Nantianqing	AAA	TR4	MR	Dongguan Banana Vegetable Institute, China	China	[51]
Dongjiao 1	AAA	TR4	MR	Dongguan Banana Vegetable Institute, China	China	[51]
Kangku 1	AAA	TR4	R	Dongguan Banana Vegetable Institute, China	China	[51]
G6-2	AAA	TR4	R	Dongguan Banana Vegetable Institute, China	China	[51]
Yueke 1	AAA	TR4	MR	Dongguan Banana Vegetable Institute, China	China	[51]
Nongke 1	AAA	TR4	MR	Dongguan Banana Vegetable Institute, China	China	[51]
Kangku 5	AAA	TR4	HR	Dongguan Banana Vegetable Institute, China	China	[51]
Nantianhuang	AAA	TR4	MR	Dongguan Banana Vegetable Institute, China	China	[51]
BXM51	AAA	TR4	MR	Dongguan Banana Vegetable Institute, China	China	[51]
Yueyoukang 1	AAA	TR4	R	South China Agricultural University	China	[113]
Pisang Gajih Merah	AAA	TR4	SS	University of Queensland, Australia	Australia	[72]
GCTCV-218 Formosana	AAA	TR4	R	University of Queensland, Australia and Northern Mozambique	China, Taiwan, Philippines and Mozambique.	[5,72]
FHIA-01 (“Goldfinger”)	AAAB	Race 1/STR4	R	DAFF, Australia [84], FHIA, Honduras [93]	Africa, Australia, Honduras	[84,93]
Tuu Gia	AA	Race 1/TR4	HR	(IFTR-GDAAS), Guangzhou, China	In breeding programs	[77]
Pisang Lilin	AA	Race 1/TR4	R	(IFTR-GDAAS), Guangzhou, China	In breeding programs	[77]
Borneo	AA	Race 1/TR4	R	National Agricultural Research Laboratories (NARL) [31] (IFTR-GDAAS), Guangzhou, China [80] and Wageningen University and Research, Wageningen, Netherlands [102]	In breeding programs	[31,77,102]
Pisang Berlin	AA	Race 1/TR4	R	(IFTR-GDAAS), Guangzhou, China [77], Embrapa cassava and fruit growing, Brazil [78]	In breeding programs	[77,78]
Zebrina GF	*	Race 1/TR4	R	University of Malaya, Kuala Lumpur, Malaysia [31], IFTR-GDAAS, Guangzhou, China [77]	In breeding programs	[31,77]
Pahang	AA	Race 1/STR4/TR4	HR	University of Queensland, Australia [72], Yunnan Agricultural University, Kunming, China [74,114] and IFTR-GDAAS, Guangzhou, China [77]	In breeding programs	[72,74,77,114]
Calcutta-4	AA	Race 1/STR4/TR4	HR	University of Queensland, Australia [66] and (IFTR-GDAAS), Guangzhou, China [72]	In breeding programs	[72,77]
Ma851	AA	STR4/TR4	R	University of Queensland, Australia	In breeding programs	[72]
Ma852	AA	STR4/TR4	R	University of Queensland, Australia	In breeding programs	[72]
Calcutta4-IV9	AA	STR4/TR4	R	University of Queensland, Australia [66] and IFTR-GDAAS, Guangzhou, China [72]	In breeding programs	[72,77]
SH-3362	AA	STR4/TR4	R	University of Queensland, Australia	In breeding programs	[72]
SH-3142	AA	STR4/TR4	R	University of Queensland, Australia	In breeding programs	[72]
Madang Guadeloupe	AA	STR4/TR4	R	University of Queensland, Australia	In breeding programs of	[72]
FHIA-1 (“Gold Finger”)	AAAB	STR4/TR4	R	University of Queensland, Australia	Australia, Brazil, Mexico, Colombia, EUA	[72]
FHIA-25	AAB	STR4/TR4	R	University of Queensland, Australia [72], (IFTR-GDAAS), Guangzhou, China [77], Wageningen University and Research, Wageningen, Netherlands [102]	Africa, Latin America and Australia (Honduras, Colombia, Brazil, Jamaica, Mozambique)	[72,77,102]
GCTCV-119	AAA	STR4/TR4	R	University of Queensland, Australia and Northern Mozambique	China, Taiwan, The Philippines, Mozambique	[5,72]
Ma850	AA	ST4/TR4	R	University of Queensland, Australia	In breeding programs	[72]
Pisang Jari Buaya	AA	STR4/TR4	R	University of Queensland, Australia [72] and (IFTR-GDAAS), Guangzhou, China [77]	In breeding programs	[72,77]
FHIA-18	AAAB	STR4/TR4	R	University of Queensland, Australia [72], IFTM Brazil [81], DAFF, Australia [84], Federal University of Santa Catarina, Brazil [111]	Africa, Latin America and Australia (Honduras, Colombia, Jamaica, Mozambique)	[72,81,84,111]

R, SS, MS, S, and HS abbreviate resistant, slightly susceptible, moderately susceptible, susceptible, and highly susceptible. EAHBs = East African Highland Bananas; IFTR-GDAAS = Institute of Fruit Tree Research, Guangdong Academy of Agricultural Sciences; EMBRAPA = Brazilian agricultural research corporation; DAFF = Department of Agriculture, Fisheries and Forestry.

**Table 5 jof-07-00249-t005:** Transcriptomic studies involving banana plants infected with *Fusarium oxysporum* f. sp. *cubense* in articles about the improvement of banana to *Fusarium* wilt, carried out in the last ten years *.

Article	Banana Variety	Plant Growth Stage	Race	Sampling (after Infection)	Pathways Enriched for Differentially Expressed Genes
Wang et al. [115]	Banana “Brazil” (susceptible) and “Formosana” (tolerant)	4.5 months	FOC TR4	48 h	Flavonoid biosynthesis, flavone and flavonol biosynthesis, alpha-linolenic acid metabolism, starch and sucrose metabolism and phenylpropanoid biosynthesis.
Wang et al. [116]	Banana “Brazil”	60 d	FOCTR4	0, 2, 4, 6 days	Phenylalanine metabolism, phenylpropanoid biosynthesis, drug metabolism—cytochrome P450, alpha-linolenic acid metabolism, amino sugar and nucleotide sugar metabolism.
Li et al. [37]	Banana “Brazil”	50 d	FOC 1 and FOC TR4	3, 27, 51 h	PR proteins, phytoalexins and phenylpropanoid synthesis, cell wall modifications, biosynthesis via ethylene signaling.
Li et al. [117]	Banana “Brazil” and “Nongke Nº 1” (resistant)	Plants with four or five leaves	FOC TR4	48, 96 h	Perception of PAMP by PRRs, hormone biosynthesis and signaling, transcription factors, cell wall modification, flavonoid biosynthesis, programmed cell death, PR proteins
Bai et al. [113]	Banana “Brazil” and “Yueyoukang 1” resistant	8 weeks (plants with five leaves)	FOC TR4	0, 5, 1, 3, 5, 10 days	PR proteins, transcription factors, cell wall modification, phenylpropanoid biosynthesis, plant hormone signal transduction.
Zhang et al. [114]	*Musa acuminata* Pahang and Brazilian		FOC STR4	at 14 days	PR proteins, transcription factors, cell wall modification, phenylpropanoid biosynthesis, plant hormone signal transduction.
Sun et al. [32]	*Musa acuminata* ” Guijiao 9” and Williams	6 months	FOC TR4	At 6 days	Membrane-bound intracellular organelle, cell wall and cytoplasm, ions, transcription factor and oxidoreductase activity, plant–pathogen interaction, plant hormone signal transduction, phenylpropanoid biosynthesis and flavonoid biosynthesis.
Fei et al. [118]	Cavendish banana	3 months	FOC 1 and FOC TR4	At 28 days	Cell components, molecular function and biological process.
Cheng et al. [103]	*Musa acuminata* cv. Tianbaojiao	11 weeks	FOC TR4	5, 10, 25 h	Auxin-activated signaling pathway, cellular response, auxin stimulation, phenylpropanoid catabolic process, lignin catabolic process, lignin metabolic process, via peroxisomes.
Song et al. [119]	Brazilian banana and señorita banana	In the five-leaf stage	FOC 1 and FOC TR4	In the five-leaf stage	Cellular process, metabolic process and binding of organelles and nucleic acids or proteins, regulation of biological processes and transcription factors.
Li et al. [120]	Cavendish banana and Brazilian (BX)	90 days	FOC TR4	27 h, 51 h	Secondary metabolite biosynthesis, plant–pathogen interaction, phenylpropanoid biosynthesis and phenylalanine metabolism, fatty acid metabolism, glycerolipid and glycerophospholipid metabolism
Niu et al. [121]	Yueyoukang1 and Baxijiao	2 weeks	FOC TR4	24 h	Cell wall biosynthesis and degradation, cell polysaccharide metabolic process, chitinase activity, pectinesterase activity and xyloglucan activity, fructose and mannose metabolism, sphingolipid metabolism, butanoate metabolism, porphyrin and chlorophyll metabolism, carotenoid and ribosome biosynthesis.

* modified table by Wang et al. [115].

**Table 6 jof-07-00249-t006:** Evaluation of hybrids and genetic inheritance studies in articles about the improvement of banana plants to fusarium wilt carried out in the last ten years.

Hybrids	Parentage
Article	
Ssali et al. [93]	
F2 progenies	Diploid TMB2X8075 (“SH3362” (AA) × “Calcutta 4” (AA) × Sukali Ndizi (AAB)
Arinaitwe et al. [31]	
F1 progenies	Monyet (*Musa acuminata* ssp. *Zebrina*) × Kokopo (*Musa acuminata* ssp. *Banksii*)
	Ahmad et al. [108]
	*Musa acuminata* ssp. *Malaccensis* (selfed)
Gonçalves et al. [33]	
CNPMF0038	((M53 × Madu)) × ((Malaccensis × Tjau Lagada))
CNPMF0496	((M61 × Pisang Lilin)) × ((Terrinha × Calcutta 4))
CNPMF0513	((M61 × Pisang Lilin)) × ((M53 × Kumburgh))
CNPMF0519	Self-fertilization (wild diploid Tambi)
CNPMF0534	((Calcutta 4 × Madang)) × ((Borneo × Guyod))
CNPMF0536	((Calcutta 4 × Madang)) × ((Borneo × Guyod))
CNPMF0542	((SH3263)) × ((Malaccensis × Sinwobogi))
CNPMF0557	((M61 × Pisang Lilin)) × ((Malaccensis × Tjau Lagada))
CNPMF0565	((Calcutta 4 × Pahang) × (Borneo × Madang)) × Khae
CNPMF0572	((Khai × (Calcutta 4 × Madang)) × ((Calcutta 4 × Madang))
CNPMF0612	((M53 × Madu) × Madu)) × SH3263
CNPMF0731	((Malaccensis × Madang)) × ((Tuugia × Calcutta 4))
CNPMF0767	((Malaccensis × Madang)) × ((Khai × (Calcutta 4 × Madang))
CNPMF0811	((Khai × (Calcutta 4 × Madang)) × ((Calcutta 4 × Pahang) × (Borneo × Madang))
CNPMF0978	((Calcutta 4 × Madang)) × ((Terrinha × Calcutta 4))
CNPMF0993	((Borneo × Guyod) × (Tuugia × Calcutta 4)) × ((Khai × (Calcutta 4 × Madang))
CNPMF0998	((Borneo × Guyod)) × ((Borneo × Guyod) × SH3263)
CNPMF1102	((Jari Buaya × (Calcutta 4 × Madang)) × ((Borneo × Guyod) × (Tuugia × Calcutta 4))
CNPMF1105	((Borneo × Guyod) × (Calcutta 4 × Heva)) × ((Calcutta 4 × Madang))
CNPMF1171	((Malaccensis × Madang)) × ((M53 × (Tuugia × Calcutta 4))
CNPMF1272	((Borneo × Guyod) × (Calcutta 4 × Heva)) × ((Tuugia × Calcutta 4))
CNPMF1286	((Calcutta 4 × Madang)) × ((Terrinha × Calcutta 4))
CNPMF1323	((Malaccensis × Sinwobogi)) × ((Calcutta 4 × Heva))
CNPMF0241	((Pacovan × improved diploid))
CNPMF0282	((Pacovan × improved diploid))
CNPMF0351	((Prata Anã × improved diploid by FHIA))
CNPMF0897	((Prata Anã)) × ((Malaccensis × Sinwobogi) × (Zebrina × Heva))
CNPMF0898	((Prata Anã)) × ((Malaccensis × Sinwobogi) × (Calcutta 4 × Galeo))
CNPMF0906	((Prata Santa Maria × improved diploid))
CNPMF0908	((Silk × improved diploid))
BRS Princesa	((Yangambi × M53))

**Table 7 jof-07-00249-t007:** Genes used transgenics in studies on genetic improvement of banana to *Fusarium* wilt in the last ten years.

Gene	Sources	Function	Banana Cultivar	References
Ferredoxin (Atfd3) and ferredoxin-like protein (pflp)	*Capsicum annuum*	Antimicrobial peptide	cv. Pei Chiao (AAA)	[69]
Petunia floral defenses (PhDef1 and PhDef2)	*Petunia hybrida*	Antimicrobial peptide	cv. Rasthali (AAB)	[124]
Onion—Ace-AMP1	*Allium cepa*	Antimicrobial peptide	cv. Rasthali (AAB)	[88]
Endochitinase (chit42)	*Trichodermaharzianum*	Antifungal activity	cv. Furenzhi (AA)	[125]
Defensin (Sm-AMP-D1)	*Stellaria media*	Antimicrobial peptide	cv. Rasthali (AAB)	[126]
Small interfering RNAs(siRNAs)/(ihpRNA)	*_*	Silencing of vital fungal genes	cv. Rasthali (AAB)	[83]
(MusaDAD1, MusaBAG1 eMusaBI1)	*Musa acuminata*	Cell death is highly induced by FOC infection	cv. Rasthali (AAB)	[96]
Cell death (Bcl-Xl, Ced-9 e Bcl-23)	*Caenorhabditiselegans*	Antiapoptosis	cv. Grand Naine	[89]
Cell death (Ced9)	*Caenorhabditiselegans*	Antiapoptosis	cv. Sukali Ndizi (*Musa* ssp. AAB)	[100]
Pathogenesis-reported (MaPR-10)	*Musa acuminata* ssp. *malaccensis*	Pathogenesis (PR)	*M. acuminata* cv. Berangan	[73]
(RGA2) and (Ced9)	*Musa acuminata* ssp. *malaccensis /Caenorhabditis elegans*	Resistance analog/antiapoptosis	cv. Grand Naine	[127]
Chitinases and 1.3-glucanase	*Oryza sativa*	Disease tolerance	cv. Rasthali (AAB)	[122]
Synthesis of ergosterol (ERG6)	*_*	Silencing of vital fungal genes	Cavendish	[128]
Small interfering RNAs–ihpRNA	*_*	Silencing of vital fungal genes	cv. Rasthali (AAB)	[129]

**Table 8 jof-07-00249-t008:** Resistance-inducing agents in banana plants reported in studies on improvement to *Fusarium* wilt in the last ten years.

Inductor	Application	References
*Bacillus subtilis*	Inoculation of plants with suspension in a greenhouse	[130]
*Trichoderma asperellum*	Inoculation of plants with suspension in a greenhouse	[131]
Abscisic acid (ABA), ethephon, methyl jasmonate (MeJA) and salicylic acid (SA)	Root treatment with inductor solutions	[132]
*Penicillium citrinum*	Inoculation of plants with suspension in a greenhouse	[88]
*Bacillus subtilis*	Treatment with in vitro fermented culture filtrate and inoculation of plants with suspension in a greenhouse	[90]
Benzothiadiazole (BTH)	Spraying leaves and roots	[133]
Interaction with dead FOC pathogen	Inoculation of plants with suspension in a greenhouse	[134]
Methyl jasmonate (MeJA)	Exogenous solution treatment in soil and leaves	[76]
A strain of FOC 1 incompatible with inducing resistance against the tropical race 4 TR4	Systemic resistance acquired by in vitro inoculation	[99]
Isolates of *Trichoderma* spp. (*T. koningii*, *T. viride*, *T. harzianum*)	Biomass, liquid culture and culture filtrate	[135]

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
