# Peer review of "Improvements in the Resistance of the Banana Species to Fusarium Wilt: A Systematic Review of Methods and Perspectives"

_jof, 2021, doi:10.3390/jof7040249_

Round 1
Reviewer 1 Report
This is a very interesting study and approach bringing together a comprehensive and systematic review of published research to highlight key targets for banana improvement in relation to control of Fusarium wilt.
The manuscript is well written and will benefit researchers working in banana improvement. Additionally the approach taken will have applications across other agricultural commodities
I have only a few comments, mostly minor edits
Line 19 and in other places (eg ln 91, 93) Musa spp. : Musa should be in italics as it is in ln 84 for example
Line 24 “this data should be “these” data. Elsewhere the authors have used the correct convention of assuming the term data is pleural. Whichever you choose, be consistent.
Line 28 “bananas” should be ”banana”. When referring to it as a crop use the singular. Again, in most cases in the manuscript the pleural has been implied but check eg lines 471, 475,480, 541 and elsewhere.
Line 61 I believe there is growing consensus that survival of Foc in the soil is not solely due to chlamydospores, that its persistence as a saprophyte or even as n endophyte on non-banana hosts is relevant.
Line 62 Although mentioned later, as part of the systemic analysis, pseudostem splitting is a consistently reported symptoms so should perhaps be mentioned here within the introduction
Line 77 long term survival of spores (comments as above relating to Line 61)
Line 77 not sure what is meant by “and culture”
Line 77 -78 Has the low efficiency of biological control been directly attributed to genetic variability and the emergence of new pathogenic strains as the statement implies.
Line 83 Several studies on what? Banana improvement? Fusarium wilt?
Line 85-87 Perhaps rearrange this sentence as it is a bit ambiguous as to why it is here. Presumably you are introducing systematic reviews as another approach and giving examples of where it has been used previously in studies unrelated to this one.
Line 102 change “who” to “which”
Line 192 is “set of packages” sufficient detail?
Line 211 in silico and in planta should be in italics and elsewhere
In Table 2 change “e” to “and” ( line 6)
Figure 6 “Transgenia”?
Line 395 Locations?
Line 403 banana plants, not “trees”
Line 440-456 In relation to this discussion point is it worth mentioning the hemibiotrophic nature of Fusarium oxysporum?
Line 468 DH Pahang
Line 515 This perhaps gets debatable as to what “resistant” actually means. In some case the term ”tolerance” is better.
Author Response
We made all the changes suggested by the reviewer I. Below the point-to-point modifications.
Comments and Suggestions for Authors
This is a very interesting study and approach bringing together a comprehensive and systematic review of published research to highlight key targets for banana improvement in relation to control of Fusarium wilt.
The manuscript is well written and will benefit researchers working in banana improvement. Additionally the approach taken will have applications across other agricultural commodities
I have only a few comments, mostly minor edits
Line 19 and in other places (eg ln 91, 93) Musa spp.: Musa should be in italics as it is in ln 84 for example - ACCEPTED
Line 24 “this data should be “these” data. Elsewhere the authors have used the correct convention of assuming the term data is pleural. Whichever you choose, be consistente - ACCEPTED
Line 28 “bananas” should be ”banana”. When referring to it as a crop use the singular. Again, in most cases in the manuscript the pleural has been implied but check eg lines 471, 475,480, 541 and elsewhere - ACCEPTED
Line 61 I believe there is growing consensus that survival of Foc in the soil is not solely due to chlamydospores, that its persistence as a saprophyte or even as n endophyte on non-banana hosts is relevant - ACCEPTED
Line 62 Although mentioned later, as part of the systemic analysis, pseudostem splitting is a consistently reported symptoms so should perhaps be mentioned here within the introduction -- ACCEPTED
Line 77 long term survival of spores (comments as above relating to Line 61) - ACCEPTED
Line 77 not sure what is meant by “and culture” - - ACCEPTED
Line 77 -78 Has the low efficiency of biological control been directly attributed to genetic variability and the emergence of new pathogenic strains as the statement implies - ACCEPTED
Line 83 Several studies on what? Banana improvement? Fusarium wilt? - ACCEPTED
Line 85-87 Perhaps rearrange this sentence as it is a bit ambiguous as to why it is here. Presumably you are introducing systematic reviews as another approach and giving examples of where it has been used previously in studies unrelated to this one - ACCEPTED
Line 102 change “who” to “which” - ACCEPTED
Line 192 is “set of packages” sufficient detail?- ACCEPTED
Line 211 in silico and in planta should be in italics and elsewhere - ACCEPTED
In Table 2 change “e” to “and” ( line 6) - ACCEPTED
Figure 6 “Transgenia”? - ACCEPTED
Line 395 Locations? - ACCEPTED
Line 403 banana plants, not “trees” - ACCEPTED
Line 440-456 In relation to this discussion point is it worth mentioning the hemibiotrophic nature of Fusarium oxysporum? - ACCEPTED
Line 468 DH Pahang - ACCEPTED
Line 515 This perhaps gets debatable as to what “resistant” actually means. In some case the term ”tolerance” is better. - ACCEPTED
Reviewer 2 Report
My only qualification for reviewing the paper is that I work with Fusarium resistance in wheat. I do not know the banana germplasm or literature so it is difficult to make specific recommendations.
The paper is well written and clearly conceived. By its nature it is descriptive rather than oriented towards hypothesis testing. That is fine but it leaves me wondering what the practical outcome of this paper will be. For example, is it likely that the call for a standardized inoculation method will be heeded? Is it likely that breeders will use the wild sources of resistance genes recommended? For this to have a practical impact it needs to be read by agricultural policy makers as well as private industry. This is not an easy problem to solve.
My suggestions are trivial:
• Capitalize FOC since it is an acronym of sorts
• Define PRISMA for those readers not familiar with it
• specify the R packages used to generate the map
Author Response
We made all the changes suggested by the reviewer II. Below the point-to-point modifications.
Comments and Suggestions for Authors
My only qualification for reviewing the paper is that I work with Fusarium resistance in wheat. I do not know the banana germplasm or literature so it is difficult to make specific recommendations.
The paper is well written and clearly conceived. By its nature it is descriptive rather than oriented towards hypothesis testing. That is fine but it leaves me wondering what the practical outcome of this paper will be. For example, is it likely that the call for a standardized inoculation method will be heeded? Is it likely that breeders will use the wild sources of resistance genes recommended? For this to have a practical impact it needs to be read by agricultural policy makers as well as private industry. This is not an easy problem to solve.
My suggestions are trivial:
- Capitalize FOC since it is an acronym of sorts - ACCEPTED
- Define PRISMA for those readers not familiar with it - ACCEPTED
- specify the R packages used to generate the map – ACCEPTED
Reviewer 3 Report
This article provides a systematic review of banana genetic improvement to Fusarium oxysporum f. sp. cubense literature published over the last decade. Systematic review software was used and, following screening and eligibility assessments, 95 articles were examined and presented. This manuscript explores 9 key research questions relating to locations of germplasm, resources and studies; sources of resistance; gene expression; breeding techniques and technologies; as well as methods for diseases assessment and data generation. It concludes that genetic improvement is the best strategy for tackling Fusarium wilt, and argues for collaboration between institutes, as well as a unified approach in plant inoculation protocols and disease assessment.
Broad Comments
This was an interesting review which addressed topics within its scope. Recommendations for a more collaborative, unified approach to Fusarium wilt resistance research are welcomed. There are some instances where the text does not match the figures. Some of the colours used in the stacked bar charts are difficult to distinguish.
Specific Comments
Introduction
- Succinct, clear introduction which flows into the manuscript.
- L40 Reword “ In African regions, plantains are cooked and consumed as an essential meal,” to “In African regions, plantains comprise a significant and essential component …”
- L54 Xanthomonas campestris pv. musacearum has recently been reclassified – see Nakato et al, 2020 Plant Pathology
- Line 60: Chlamydospores are referred to as “resistant”, I would be cautious using this term. Perhaps better to use the term “resting”.
- Additionally, the paragraph beginning on line 57 might also consider/mention alternative hosts enabling Foc to persist in production areas.
- Line 69: Correct to “A highly virulent strain of Foc able to infect Cavendish cultivars…”, not a “a highly virulent strain of Cavendish cultivars was identified…”
- L 74 Reword “The low efficiency of biological control is attributed to factors inherent to the primary inoculum dynamics of the disease, primarily the long-term survival of spores in soil and culture remains, and to the genetic variability of the pathogen, resulting in new strains capable of infecting resistant cultivars”
Materials and Methods
- Clear, easy to follow.
- Did the authors consider the use of new species names for Foc proposed by Maryani et al., (2019) in their standardised search string?
Maryani, N., L. Lombard, Y. S. Poerba, S. Subandiyah, P. W. Crous, and G. H. J. Kema. "Phylogeny and genetic diversity of the banana Fusarium wilt pathogen Fusarium oxysporum f. sp. cubense in the Indonesian centre of origin." Studies in Mycology 92 (2019): 155-194.
Results
- L 154 What is CAPES?
- Section 3.1 Screening of Results – it may be beneficial to have a supplementary table listing which articles were from which sources. I am keen to know which articles were added manually, particularly as this may indicate how effective the standardised search string is.
- Figure 4: What are the R packages used to generate the image?
- All stacked bar charts: Colours challenging to distinguish.
- Figure 5B: Why are glasshouse and greenhouse treated as separate, but terms used interchangeably? In text: Vegetation houses (13%), greenhouses (2%). In figure: greenhouse (12.6%), glasshouse (2.1%). Should these not be categorised together?
- P10 - need to define RAPD, SCAR
- Figure 10B/line 295: in the text the value of 2% is given for categories given 8% in the figure. I believe the 2% is incorrect.
- Table 4: There appear to be random occurrences of the letter “e”. Please clarify its use or remove it.
- Line 315: Capitalise Fusarium.
- L 348 “bio-ballistics” - change to “biolistics”
Discussion
- Overall a strong discussion, with most points considered and clearly explained.
- Line 467: Italicise itinerans
- L517 - -what does “vegetable hormones” mean?
- The suggestion for a universal infection protocol and symptom analysis is sensible.
- The use of CRISPR/Cas9 is mentioned as a tool which is likely to be used in the future. There is an article on this from the Tripathi lab in last years current Opinions in Plant Biology. Suggest the authors also includethe use of resistance gene enrichment sequencing (RenSeq) and its potential applications.
Conclusions
- Nicely concluded.
- Line 596: there appears to be a space between “CRISPR/Cas” and “9”.
- L 564 - remove “a”
References
- Line 780: space between “disease” and “and”
Author Response
We made all the changes suggested by the reviewer III. Below the point-to-point modifications.
Comments and Suggestions for Authors
This article provides a systematic review of banana genetic improvement to Fusarium oxysporum f. sp. cubense literature published over the last decade. Systematic review software was used and, following screening and eligibility assessments, 95 articles were examined and presented. This manuscript explores 9 key research questions relating to locations of germplasm, resources and studies; sources of resistance; gene expression; breeding techniques and technologies; as well as methods for diseases assessment and data generation. It concludes that genetic improvement is the best strategy for tackling Fusarium wilt, and argues for collaboration between institutes, as well as a unified approach in plant inoculation protocols and disease assessment.
Broad Comments
This was an interesting review which addressed topics within its scope. Recommendations for a more collaborative, unified approach to Fusarium wilt resistance research are welcomed. There are some instances where the text does not match the figures. Some of the colours used in the stacked bar charts are difficult to distinguish - ACCEPTED
Specific Comments
Introduction
- Succinct, clear introduction which flows into the manuscript
- L40 Reword “ In African regions, plantains are cooked and consumed as an essential meal,” to “In African regions, plantains comprise a significant and essential component …” - ACCEPTED
- L54 Xanthomonas campestris pv. musacearumhas recently been reclassified – see Nakato et al, 2020 Plant Pathology - ACCEPTED
- Line 60: Chlamydospores are referred to as “resistant”, I would be cautious using this term. Perhaps better to use the term “resting” - ACCEPTED
- Additionally, the paragraph beginning on line 57 might also consider/mention alternative hosts enabling Foc to persist in production areas. - - ACCEPTED
- Line 69: Correct to “A highly virulent strain of Foc able to infect Cavendish cultivars…”, not a “a highly virulent strain of Cavendish cultivars was identified…”- ACCEPTED
- L 74 Reword “The low efficiency of biological control is attributed to factors inherent to the primary inoculum dynamics of the disease, primarily the long-term survival of spores in soil and culture remains, and to the genetic variability of the pathogen, resulting in new strains capable of infecting resistant cultivars” - ACCEPTED AND WRITTEN AGAIN
Materials and Methods
Clear, easy to follow.
- Did the authors consider the use of new species names for Foc proposed by Maryani et al., (2019) in their standardised search string? Não consideramos, nem mencionamos essa classificação em nossa revisão. ACCEPTED: We inserted the new nomenclature in the search string to test and found that there was a very large restriction of articles within the search period for some databases, for example, the academic google which from 377 articles started to present 13 articles. In other databases, such as PubMed Central, this change increased the number of articles in relation to what we had found, but the vast majority was outside the proposed theme. We used the string as follows, in our test: Musa spp. and bananas and plantains and Fusarium wilt or Fusarium oxysporum f. sp. cubense or Fusarium odoratissimum or Panama disease and genetic resistance and markers and genes.
Maryani, N., L. Lombard, Y. S. Poerba, S. Subandiyah, P. W. Crous, and G. H. J. Kema. "Phylogeny and genetic diversity of the banana Fusarium wilt pathogen Fusarium oxysporum f. sp. cubense in the Indonesian centre of origin." Studies in Mycology 92 (2019): 155-194.
Results
- L 154 What is CAPES? - ACCEPTED
- Section 3.1 Screening of Results – it may be beneficial to have a supplementary table listing which articles were from which sources. I am keen to know which articles were added manually, particularly as this may indicate how effective the standardised search string is - ACCEPTED
- Figure 4: What are the R packages used to generate the image? - ACCEPTED
- All stacked bar charts: Colours challenging to distinguish - ACCEPTED
- Figure 5B: Why are glasshouse and greenhouse treated as separate, but terms used interchangeably? In text: Vegetation houses (13%), greenhouses (2%). In figure: greenhouse (12.6%), glasshouse (2.1%). Should these not be categorised together? – ACCEPTED: We corrected the text and chose to keep the categories separate.
- P10 - need to define RAPD, SCAR - ACCEPTED
- Figure 10B/line 295: in the text the value of 2% is given for categories given 8% in the figure. I believe the 2% is incorrect - ACCEPTED
- Table 4: There appear to be random occurrences of the letter “e”. Please clarify its use or remove it - ACCEPTED
- Line 315: Capitalise Fusarium - ACCEPTED
- L 348 “bio-ballistics” - change to “biolistics” - ACCEPTED
Discussion
- Overall a strong discussion, with most points considered and clearly explained.
- Line 467: Italicise itinerans - ACCEPTED
- L517 - -what does “vegetable hormones” mean? - ACCEPTED
- The suggestion for a universal infection protocol and symptom analysis is sensible.
- The use of CRISPR/Cas9 is mentioned as a tool which is likely to be used in the future. There is an article on this from the Tripathi lab in last years current Opinions in Plant Biology. Suggest the authors also includethe use of resistance gene enrichment sequencing (RenSeq) and its potential applications. – - ACCEPTED
Conclusions
- Nicely concluded.
- Line 596: there appears to be a space between “CRISPR/Cas” and “9”.- ACCEPTED
- L 564 - remove “a” - ACCEPTED
References
- Line 780: space between “disease” and “and” - ACCEPTED
We attached the following files, after corrections: Manuscript; Figures 2, 5, 6, 8, 9 and 10; Table 2, 3 and 4; Supplementary Table 3.